# Non-invasive detection of urothelial cancer through the analysis of driver gene mutations and aneuploidy

Simeon U Springer[1,2†], Chung-Hsin Chen[3†], Maria Del Carmen Rodriguez Pena[4,5†], Lu Li[6], Christopher Douville[7], Yuxuan Wang[1,2], Joshua David Cohen[1,2], Diana Taheri[4,8], Natalie Silliman[1,2], Joy Schaefer[1,2], Janine Ptak[1,2], Lisa Dobbyn[1,2], Maria Papoli[1,2], Isaac Kinde[1,2], Bahman Afsari[9,10], Aline C Tregnago[4], Stephania M Bezerra[11], Christopher VandenBussche[4], Kazutoshi Fujita[12], Dilek Ertoy[13], Isabela W Cunha[11], Lijia Yu[5], Trinity J Bivalacqua[14], Arthur P Grollman[15,16], Luis A Diaz[17], Rachel Karchin[7,9], Ludmila Danilova[10,13], Chao-Yuan Huang[3], Chia-Tung Shun[18], Robert J Turesky[19,20], Byeong Hwa Yun[19,20], Thomas A Rosenquist[15], Yeong-Shiau Pu[3], Ralph H Hruban[4], Cristian Tomasetti[6,10], Nickolas Papadopoulos[1,2], Ken W Kinzler[1,2], Bert Vogelstein[1,2*], Kathleen G Dickman[15,16*], George J Netto[4,5*]

[1]Howard Hughes Medical Institute, Ludwig Center for Cancer Genetics and Therapeutics, Baltimore, United States; [2]Sidney Kimmel Comprehensive Cancer Center, Baltimore, United States; [3]Department of Urology, National Taiwan University Hospital, Taipei, Taiwan; [4]Department of Pathology, Johns Hopkins University, Baltimore, United States; [5]Department of Pathology, University of Alabama at Birmingham, Birmingham, United States; [6]Department of Biostatistics, Johns Hopkins Bloomberg School of Public Health, Baltimore, United States; [7]Department of Biomedical Engineering, Institute for Computational Medicine, Johns Hopkins University, Baltimore, United States; [8]Department of Pathology, Isfahan Kidney Diseases Research Center, Isfahan University of Medical Sciences, Isfahan, Iran; [9]Department of Oncology, Johns Hopkins University, Baltimore, United States; [10]Division of Biostatistics and Bioinformatics, Department of Oncology, Sidney Kimmel Cancer Center, Johns Hopkins School of Medicine, Baltimore, United States; [11]Department of Pathology, AC Camargo Cancer Center, Sao Paulo, Brazil; [12]Department of Pathology, Osaka University, Osaka, Japan; [13]Department of Pathology, Hacettepe University, Ankara, Turkey; [14]Department of Urology, Johns Hopkins University, Baltimore, United States; [15]Department of Pharmacological Sciences, Stony Brook University, Stony Brook, United States; [16]Department of Medicine, Stony Brook University, Stony Brook, United States; [17]Department of Medicine, Memorial Sloan Kettering Cancer Center, New York, United States; [18]Department of Forensic Medicine and Pathology, National Taiwan University Hospital, Taipei, Taiwan; [19]Masonic Cancer Center, University of Minnesota, Minneapolis, United States; [20]Department of Medicinal Chemistry, University of Minnesota, Minneapolis, United States

*For correspondence:
bertvog@gmail.com (BV);
kathleen.dickman@stonybrook.edu (KGD);
gnetto@uabmc.edu (GJN)

†These authors contributed equally to this work

**Abstract** Current non-invasive approaches for detection of urothelial cancers are suboptimal. We developed a test to detect urothelial neoplasms using DNA recovered from cells shed into urine. UroSEEK incorporates massive parallel sequencing assays for mutations in 11 genes and

copy number changes on 39 chromosome arms. In 570 patients at risk for bladder cancer (BC), UroSEEK was positive in 83% of those who developed BC. Combined with cytology, UroSEEK detected 95% of patients who developed BC. Of 56 patients with upper tract urothelial cancer, 75% tested positive by UroSEEK, including 79% of those with non-invasive tumors. UroSEEK detected genetic abnormalities in 68% of urines obtained from BC patients under surveillance who demonstrated clinical evidence of recurrence. The advantages of UroSEEK over cytology were evident in low-grade BCs; UroSEEK detected 67% of cases whereas cytology detected none. These results establish the foundation for a new non-invasive approach for detection of urothelial cancer.
DOI: https://doi.org/10.7554/eLife.32143.001

## Introduction

Bladder cancer (BC) is the most common malignancy of the urinary tract. According to the American Cancer Society, 79,030 new cases of bladder cancer and 18,540 deaths were estimated to occur in the United States alone in 2017 (*Siegel et al., 2017*). Predominantly of urothelial histology, invasive BC arises from non-invasive papillary or flat precursors, and many BC patients suffer multiple relapses prior to progression, providing ample lead-time for early detection and treatment prior to metastasis (*Netto, 2013*).

Although most urothelial carcinomas arise in the bladder, 5–10% originate in the renal pelvis and/or ureter (*Rouprêt et al., 2015*; *Soria et al., 2017*). The annual incidence of these upper tract urothelial carcinomas (UTUC) in Western countries is 1–2 cases per 100,000 (*Rouprêt et al., 2015*; *Soria et al., 2017*), but the disease occurs at a much higher rate in populations exposed to aristolochic acid (AA) (*Chen et al., 2012*; *Grollman, 2013*; *Lai et al., 2010*; *Taiwan Cancer Registry, 2017*). AA is a carcinogenic and nephrotoxic nitrophenanthrene carboxylic acid produced by *Aristolochia* plants (*National Toxicology Program, 2011*). An etiological link between AA exposure and UTUC has been established in several populations (*Grollman, 2013*; *Grollman et al., 2009*; *Jelaković et al., 2012*; *National Toxicology Program, 2011*). The profound public health threat posed by the medicinal use of *Aristolochia* herbs is illustrated in Taiwan, which has the highest incidence of UTUC in the world (*Chen et al., 2012*; *Yang et al., 2002*). In recent years, more than one-third of the adult population in Taiwan has been prescribed herbal remedies containing AA (*Hsieh et al., 2008*), resulting in an unusually high (37%) proportion of UTUC cases relative to urothelial cancers worldwide (*Taiwan Cancer Registry, 2017*).

Tumors of the upper and lower urinary tracts differ in important ways, including etiology, but they have many common features (*Green et al., 2013*), such as the somatic alterations that drive their growth (*Lee et al., 2018*). High rates of activating mutations in the upstream promoter of the *TERT* gene are found in the majority of urothelial neoplasms of both upper and lower tracts (*Huang et al., 2013*; *Killela et al., 2013*; *Scott et al., 2014*; *Yuan et al., 2016*). *TERT* promoter mutations predominantly affect two hot spots, g.1295228 C > T and g.1295250 C > T. These mutations generate CCGGAA/T or GGAA/T motifs that alter the binding site for ETS transcription factors and subsequently stimulate increased *TERT* promoter activity (*Horn et al., 2013*; *Huang et al., 2013*). *TERT* promoter mutations occur in up to 80% of invasive urothelial carcinomas of the bladder and in several of their histologic variants (*Allory et al., 2014*; *Cowan et al., 2016*; *Killela et al., 2013*; *Kinde et al., 2013*; *Nguyen et al., 2016*). Moreover, *TERT* promoter mutations occur in 60–80% of BC precursors, including papillary urothelial neoplasms of low malignant potential (*Rodriguez Pena et al., 2017*), non-invasive low-grade papillary urothelial carcinoma, non-invasive high-grade papillary urothelial carcinoma and 'flat' carcinoma in situ (CIS), as well as in urinary cells from a subset of these patients (*Kinde et al., 2013*). *TERT* promoter mutations have thus been established as a common genetic alteration in urothelial neoplasms (*Cheng et al., 2017*; *Killela et al., 2013*; *Kinde et al., 2013*; *Yuan et al., 2016*).

Other important oncogene-activating mutations include those in *FGFR3*, *RAS* and *PIK3CA* genes, which occur in a high fraction of non-muscle invasive bladder cancers (*Humphrey et al., 2016*; *Netto, 2011*). In muscle-invasive bladder cancers, mutations in the *TP53*, *CDKN2A*, *MLL* and *ERBB2* genes are also frequently found (*Cancer Genome Atlas Research Network, 2014*; *Lin et al., 2010*; *Mo et al., 2007*; *Netto, 2011*; *Sarkis et al., 1995*; *Sarkis et al., 1994*; *Sarkis et al., 1993*;

*Wu, 2005*). Mutations in these genes are also present in UTUC (*Hoang et al., 2013*; *Lee et al., 2018*; *Moss et al., 2017*; *Sfakianos et al., 2015*).

Urine cytology is a non-invasive method for the detection of BC. Although it has value for the detection of high-grade BC, the test is unable to detect the vast majority of low-grade tumors (*Lotan and Roehrborn, 2003*; *Netto and Tafe, 2016*; *Zhang et al., 2016*). Urine cytology also fails to detect the majority of UTUCs (*Baard et al., 2017*). These facts, together with the high cost and invasive nature of repeated endoscopy and follow-up biopsy procedures, have led to many attempts to develop alternative minimally invasive methods to detect urothelial cancers. Strategies to identify BC include urine- or serum-based genetic and protein assays for screening and surveillance (*Allory et al., 2014*; *Bansal et al., 2014*; *Ellinger et al., 2015*; *Fradet and Lockhard, 1997*; *Goodison et al., 2012*; *Hurst et al., 2014*; *Kawauchi et al., 2009*; *Kinde et al., 2013*; *Krüger et al., 2003*; *Moonen et al., 2007*; *Ralla et al., 2014*; *Sarosdy et al., 2006*; *Serizawa et al., 2011*; *Skacel et al., 2003*; *Wang et al., 2014*; *Yafi et al., 2015*). Currently available U.S. Food and Drug Administration (FDA) approved assays for BC diagnosis include the ImmunoCyt test (Scimedx Corp), the nuclear matrix protein 22 (NMP22) immunoassay test (Matritech), and multitarget FISH (UroVysion) (*Fradet and Lockhard, 1997*; *Kawauchi et al., 2009*; *Krüger et al., 2003*; *Moonen et al., 2007*; *Sarosdy et al., 2006*; *Skacel et al., 2003*; *Yafi et al., 2015*). Sensitivities between 62% and 69% and specificities between 79% and 89% have been reported for some of these tests. However, due to assay performance inconsistencies, cost or requirements for technical expertise, integration of such assays into routine clinical practice has not yet occurred. Furthermore, none of the FDA-approved BC assays have yet been validated for clinical use in detection of UTUC. Therefore, a non-invasive test that predicts which patients are most likely to develop urothelial cancer could be medically and economically important.

As urothelial cells from the upper and lower urinary tracts are in direct contact with urine, we hypothesized that genetic analyses of exfoliated urinary cells could be used to detect neoplasia in these organs in a non-invasive fashion. The current study assesses the performance of a massively parallel sequencing-based assay, termed UroSEEK, for the detection of BC and UTUC through the genetic analysis of urinary cell DNA. UroSEEK has three components: detection of intragenic mutations in regions of ten genes (*FGFR3, TP53, CDKN2A, ERBB2, HRAS, KRAS, PIK3CA, MET, VHL* and *MLL*) frequently mutated in urothelial tumors (*Cancer Genome Atlas Research Network, 2014*; *Hoang et al., 2013*; *Lee et al., 2018*; *Lin et al., 2010*; *Mo et al., 2007*; *Moss et al., 2017*; *Netto, 2011*; *Sarkis et al., 1995*; *Sarkis et al., 1994*; *Sarkis et al., 1993*; *Sfakianos et al., 2015*; *Wu, 2005*); detection of mutations in the *TERT* promoter (*Huang et al., 2013*; *Killela et al., 2013*; *Scott et al., 2014*; *Yuan et al., 2016*); and detection of aneuploidy (*Kinde et al., 2012*; *Kinde et al., 2011*). The exons and specific genes chosen for inclusion in UroSEEK were chosen on the basis of BC mutations recorded in the COSMIC database (*Supplementary file 1*). Selected amplicons from VHL and MET were also included in the hope that renal cell carcinomas might also shed cells into urine, although urine samples from patients with these cancers were not included in our study.

UroSEEK was applied to three independent cohorts of patients. The first (called the BC early detection cohort) exhibited microscopic hematuria or dysuria and supplied urine samples prior to any surgical procedures. A small fraction (4% to 5%) of patients with microscopic hematuria later develops urothelial malignancy (*Wein et al., 2012*; *Mishriki et al., 2008*), so the decision as to which patients should undergo cystoscopy is often difficult. The second cohort (called the UTUC cohort) consisted of Taiwanese patients with UTUC who supplied a urine sample prior to nephroureterectomy. Such patients might benefit from a non-invasive test that could be used to screen individuals at increased risk for UTUC, such as those exposed to herbal remedies containing the carcinogen AA. The third cohort (called the BC surveillance cohort) included patients who had already been diagnosed with BC and were therefore at high risk for recurrence (*Wein et al., 2012*). Because urine cytology is relatively insensitive for the detection of recurrence, cystoscopies are performed as often as every three months in such patients in the U.S. In fact, the cost of managing these patients is in aggregate higher than the cost of managing any other type of cancer, amounting to 3 billion dollars annually (*Netto and Epstein, 2010*).

## Results

A schematic of the approach used in this study is provided in *Figure 1*. A flow diagram indicating the number of patients evaluated in this study and the major results are presented in *Figure 2*.

### BC early detection cohort
#### Cohort characteristics

A total of 570 patients were included in the early detection cohort, each with one urine sample analyzed. 90% of the patients had hematuria, 3% had lower urinary tract symptoms (LUTS), and 9% had other indications suggesting they were at risk for BC. The median age of the participants was 58 years (range 5 to 89 years; *Table 1a*). As expected from prior studies of patients at risk for BC, 70%

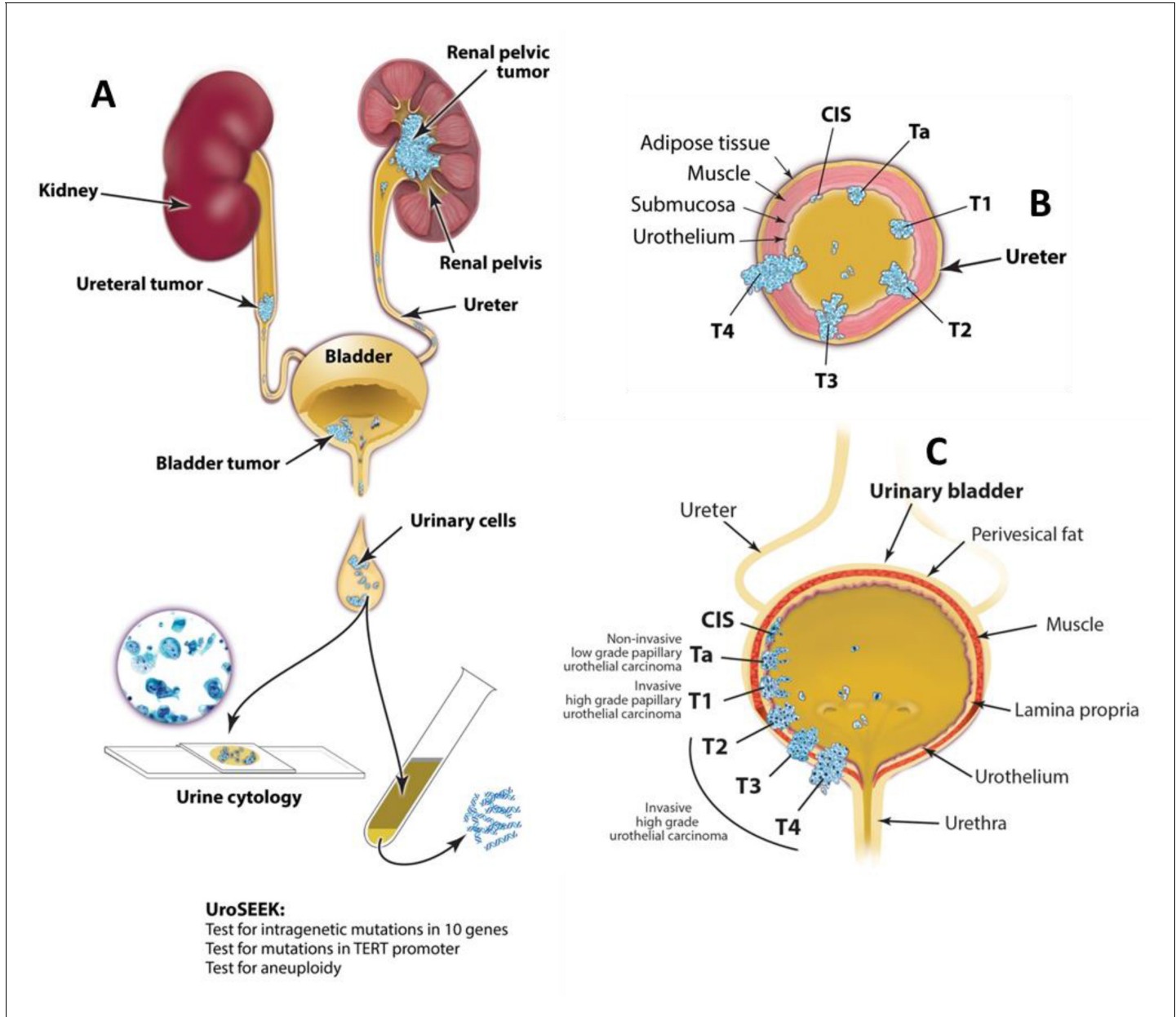

**Figure 1.** Schematic of the approach used to evaluate urinary cells in this study. UroSEEK assay is designed to detect urothelial neoplasms that are in direct contact with urine (A) of variable pathologic stages originating in upper urinary tract (B) or bladder (C).
DOI: https://doi.org/10.7554/eLife.32143.002

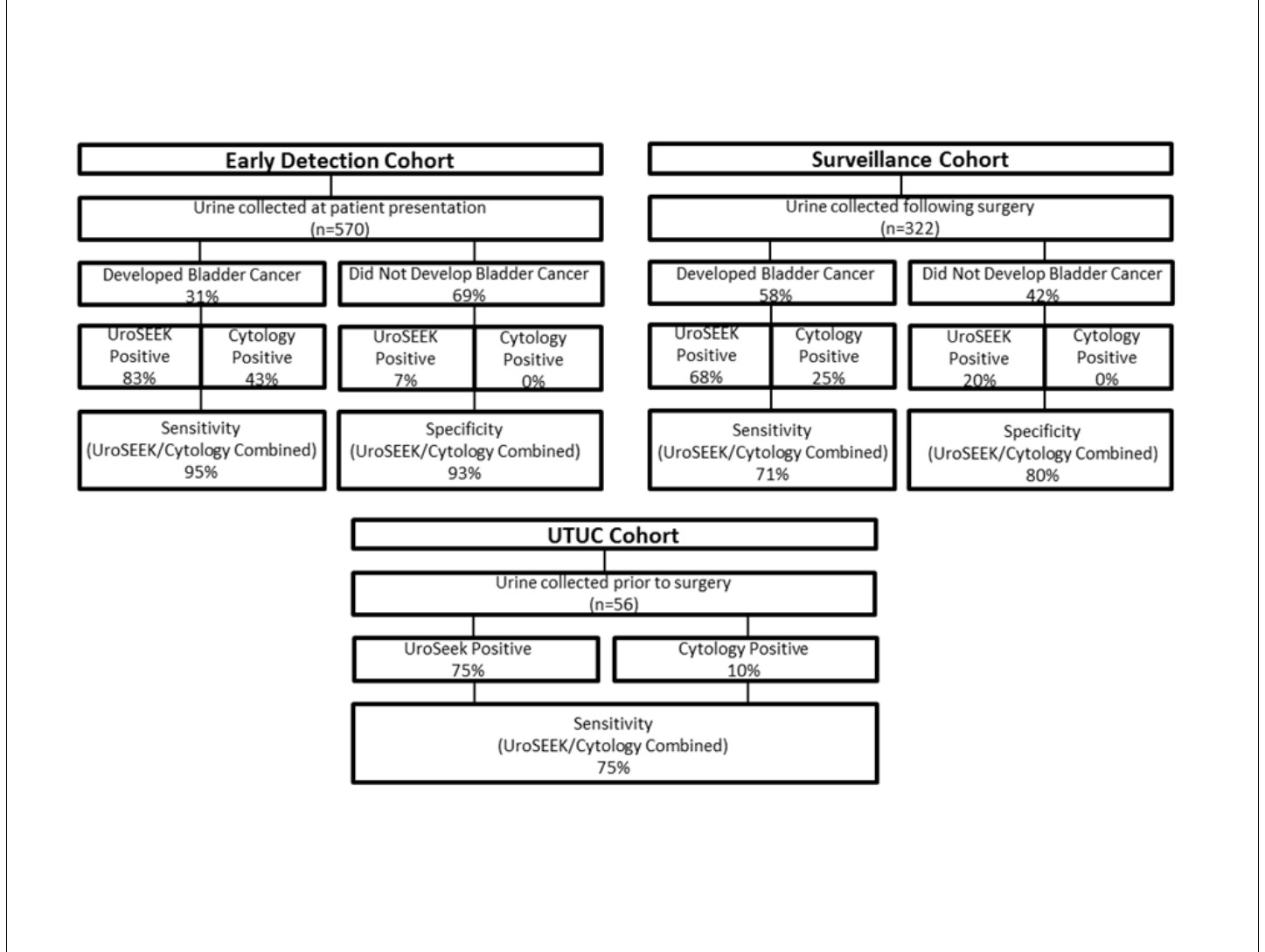

**Figure 2.** Flow diagram indicating the number of patients in the three cohorts evaluated in this study and summarizing the salient findings. Cytology was performed on only a subset of the patients (see main text).

DOI: https://doi.org/10.7554/eLife.32143.003

of the patients were male (*Siegel et al., 2017*; *Wein et al., 2012*). Patients (*n* = 175; 31%) developed BC after a median follow-up period of 18 months (range 0 to 40 months). For each patient who developed BC, we selected two other patients who presented with similar symptoms but did not develop BC during the follow-up period. By design, the fraction of cases in this cohort developing BC was higher than the fraction (5%) of patients with similar presentations who would have developed BC in standard clinical practice. The characteristics of the tumors developing in the 570 patients are summarized in *Table 1a* and detailed in *Supplementary file 2*.

## Genetic analysis

We performed three separate tests for genetic abnormalities that might be found in urinary cells derived from BC (*Figure 1*). First, we evaluated mutations in selected regions of ten genes that have been shown to be frequently altered in urothelial tumors (*Figure 3* and *Supplementary file 3*). For this purpose, we designed a specific set of primers that allowed us to detect mutations in as few as 0.03% of urinary cells (*Supplementary file 4*). The capacity to detect such low-mutant fractions was a result of the incorporation of molecular barcodes in each of the primers, thereby substantially

**Table 1.** Demographic, clinical and genetic features of the early detection cohort.

| Gender | n | % | Ten-gene multiplex positive | TERT positive | Aneuploidy positive | UroSEEK positive | Cytology positive* | Uroseek or cytology positive* |
|---|---|---|---|---|---|---|---|---|
| Table 1a. Demographic, clinical and genetic features of the early detection cohort | | | | | | | | |
| Males without recurrence | 172 | 59% | 3 (2%) | 10 (6%) | 2 (1%) | 13 (8%) | 0 (0%) | 13 (8%) |
| Males with recurrence | 32 | 11% | 26 (81%) | 21 (66%) | 19 (59%) | 29 (91%) | 16 (50%) | 30 (94%) |
| Females without recurrence | 81 | 28% | 2 (2%) | 2 (2%) | 1 (1%) | 5 (6%) | 0 (0%) | 5 (6%) |
| Females with recurrence | 9 | 3% | 4 (44%) | 4 (44%) | 3 (33%) | 6 (67%) | 1 (11%) | 6 (67%) |
| *Indication* | | | | | | | | |
| Hematuria without recurrence | 346 | 61% | 6 (2%) | 15 (4%) | 5 (1%) | 22 (6%) | 0 (0%) | 17 (5%) |
| Hematuria with recurrence | 163 | 29% | 108 (66%) | 90 (55%) | 76 (47%) | 134 (82%) | 18 (11%) | 32 (2%) |
| LUTS without recurrence | 11 | 2% | 0 (0%) | 2 (18%) | 0 (0%) | 2 (18%) | 0 (0%) | 2 (18%) |
| LUTS with recurrence | 3 | 1% | 2 (67%) | 1 (33%) | 0 (0%) | 2 (67%) | 1 (33%) | 2 (67%) |
| Other without recurrence | 38 | 7% | 1 (3%) | 0 (0%) | 1 (3%) | 2 (5%) | 0 (0%) | 2 (5%) |
| Other with recurrence | 9 | 2% | 9 (100%) | 8 (89%) | 5 (56%) | 9 (100%) | 2 (22%) | 9 (100%) |
| *Detected Tumor Diagnosis* | | | | | | | | |
| PUNLMP | 2 | 1% | 0 (0%) | 1 (50%) | 0 (0%) | 1 (50%) | 0 (0%) | 0 (0%) |
| CIS | 7 | 5% | 4 (57%) | 4 (57%) | 1 (14%) | 6 (86%) | 3 (43%) | 6 (86%) |
| LGTCC | 31 | 21% | 15 (48%) | 18 (58%) | 9 (29%) | 22 (71%) | 0 (0%) | 4 (13%) |
| HGTCC | 49 | 33% | 34 (69%) | 28 (57%) | 26 (53%) | 40 (82%) | 4 (8%) | 11 (22%) |
| INTCC | 61 | 41% | 48 (79%) | 36 (59%) | 35 (57%) | 57 (93%) | 9 (15%) | 16 (26%) |
| *Cytology diagnosis** | | | | | | | | |
| Positive | 21 | 6% | 16 (76%) | 12 (57%) | 16 (76%) | 20 (95%) | N/A | N/A |
| Atypical | 105 | 30% | 21 (20%) | 21 (30%) | 12 (11%) | 30 (29%) | N/A | N/A |
| Negative | 221 | 64% | 4 (2%) | 9 (4%) | 1 (0.4%) | 12 (5%) | N/A | N/A |
| Table 1b. Demographic, clinical and genetic features of the Surveillance cohort. | | | | | | | | |
| *Males without recurrence* | 59 | 30% | 3 (5%) | 8 (14%) | 3 (5%) | 10 (17%) | 0 (0%) | 8 (14%) |
| *Males with recurrence* | 90 | 45% | 45 (50%) | 53 (59%) | 20 (22%) | 59 (66%) | 20 (22%) | 53 (59%) |
| *Females without recurrence* | 17 | 9% | 5 (29%) | 3 (18%) | 0 (0%) | 6 (35%) | 0 (0%) | 6 (35%) |
| *Females with recurrence* | 33 | 17% | 15 (45%) | 19 (58%) | 11 (33%) | 33 (100%) | 6 (18%) | 19 (58%) |
| *Original Tumor Diagnosis* | | | | | | | | |
| *PUNLMP* | 12 | 4% | 5 (42%) | 2 (17%) | 1 (8%) | 6 (50%) | 0 (0%) | 2 (17%) |
| *CIS* | 25 | 8% | 11 (44%) | 13 (52%) | 6 (24%) | 14 (56%) | 5 (20%) | 10 (40%) |
| *LGTCC* | 107 | 35% | 27 (25%) | 34 (32%) | 8 (7%) | 41 (38%) | 0 (0%) | 59 (55%) |
| *HGTCC* | 62 | 20% | 22 (36%) | 24 (39%) | 10 (16%) | 30 (49%) | 4 (7%) | 16 (26%) |
| *INTCC* | 104 | 34% | 39 (38%) | 47 (45%) | 29 (28%) | 54 (52%) | 20 (19%) | 34 (33%) |
| *Original Tumor Stage* | | | | | | | | |
| *pTis* | 25 | 8% | 11 (44%) | 13 (52%) | 6 (24%) | 14 (56%) | 5 (20%) | 10 (40%) |
| *pTa* | 181 | 58% | 54 (30%) | 60 (33%) | 19 (19%) | 77 (43%) | 4 (2%) | 77 (43%) |

*Table 1 continued on next page*

*Table 1 continued*

| Gender | n | % | Ten-gene multiplex positive | TERT positive | Aneuploidy positive | UroSEEK positive | Cytology positive* | Uroseek or cytology positive* |
|---|---|---|---|---|---|---|---|---|
| pT1 | 71 | 23% | 28 (39%) | 35 (49%) | 22 (31%) | 39 (55%) | 14 (20%) | 23 (32%) |
| pT2 | 23 | 7% | 9 (9%) | 9 (39%) | 7 (30%) | 12 (52%) | 5 (22%) | 10 (43%) |
| pT3 | 9 | 3% | 1 (11%) | 2 (22%) | 0 | 2 (22%) | 1 (11%) | 1 (11%) |
| pT4 | 1 | 0.3% | 1 (100%) | 1 (100%) | 0 | 1 (100%) | N/A | N/A |
| *Routine cytology diagnosis** | | | | | | | | |
| Positive | 30 | 15% | 21 (21%) | 25 (83%) | 20 (67%) | 27 (90%) | N/A | N/A |
| Atypical | 95 | 48% | 38 (40%) | 43 (45%) | 18 (19%) | 50 (53%) | N/A | N/A |
| Negative | 71 | 36% | 12 (17%) | 13 (18%) | 3 (4%) | 19 (27%) | N/A | N/A |

*Cytology was available on only a subset of cases.

N/A Not Available.

DOI: https://doi.org/10.7554/eLife.32143.004

reducing the artifacts associated with massively parallel sequencing (*Kinde et al., 2011*). Second, we evaluated *TERT* promoter mutations. A singleplex PCR was used for this analysis because the unusually high GC-content of the *TERT* promoter precluded its inclusion in the multiplex PCR design. Third, we evaluated the extent of aneuploidy using a technique in which a single PCR is used to co-amplify ~38,000 members of a subfamily of long interspersed nucleotide element-1 retrotransposons (L1 retrotransposons, also called LINEs). L1 retrotransposons, like other human repeats, have spread

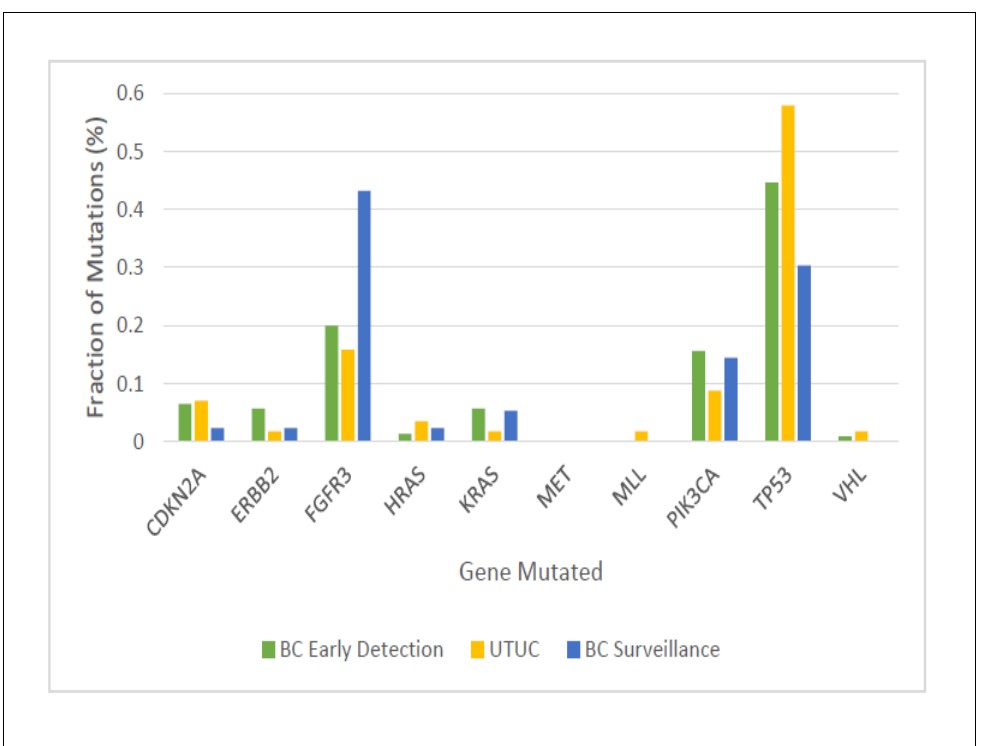

**Figure 3.** Fraction of mutations found in the ten-gene panel in 231 urinary cell samples assessed in the BC early detection cohort, 56 urinary cell samples assessed in the UTUC cohort, and 132 urinary cell samples assessed in the BC surveillance cohort.

DOI: https://doi.org/10.7554/eLife.32143.005

throughout the genome via retrotransposition and are found on all 39 non-acrocentric autosomal arms (*Kinde et al., 2012*).

The multiplex assay detected mutations in 68% of the 175 urinary cell samples from the individuals who developed BC during the course of this study (95% CI, 61% to 75%) (*Table 1a* and *Supplementary file 5*). A total of 246 mutations were detected in eight of the ten target genes (*Figure 3* and *Supplementary file 5*). The median mutant allele frequency (MAF) in the urinary cells with detectable mutations was 8% (8.14%. The most commonly altered genes were *TP53* (45% of the total mutations) and *FGFR3* (20% of the total mutations; *Figure 3*). The distribution of mutant genes was roughly consistent with expectations based on previous exome-wide sequencing studies of BCs (*Cancer Genome Atlas Research Network, 2014*). At the thresholds used, 1.7% of the 395 patients in the early detection cohort who did not develop BC during the course of the study had a detectable mutation in any of the ten genes. At the same thresholds, none of the 188 urinary cell samples from healthy individuals had a mutation in any of the ten genes assayed (100% specificity, 95% CI, 98% to 100%).

Mutations in the *TERT* promoter were detected in 57% of the 175 urinary cell samples from the patients who developed cancer during the study interval (95% CI, 49% to 64%; *Table 1a* and *Supplementary file 6*). The median *TERT* MAF in the urinary cells was 6% (5.76%). Mutations were detected in three positions; 98% of the mutations were at TERT:g.1295228 (79%) and TERT:g.1295250 (19%), which are 66 and 88 bp upstream of the transcription start site, respectively. The remaining 2% of mutations were found at TERT:g.1295242. The first two of these positions have been previously shown to be critical for the appropriate transcriptional regulation of *TERT*. In particular, the mutant alleles recruit the GABPA/B1 transcription factor, resulting in the H3K4me2/3 mark of active chromatin and reversing the epigenetic silencing present in normal cells (*Stern et al., 2015*). Of the 395 patients in this cohort who did not develop BC during the course of the study, only 4% had a detectable mutation in the *TERT* promoter. Finally, only one of the 188 urinary samples from healthy individuals harbored a *TERT* promoter mutation.

Aneuploidy was detected in 46% (95% CI, 39% to 54%) of the 175 urinary cell samples from the patients who developed BC during the course of the study (*Table 1a* and *Supplementary file 7*). The most commonly altered chromosome arms were 5q, 8q, and 9p. These three chromosome arms harbor well-known oncogenes and tumor suppressor genes that have been shown to undergo copy number alterations in many cancers, including BC (*Vogelstein et al., 2013*). Aneuploidy was detected in 1.5% of the urinary cell samples from the 395 patients who did not develop BC during the course of the study, but it was not detected in any of the 188 urinary samples from healthy individuals.

## Comparison with primary tumors

DNAs from resected or biopsied tumor samples from 102 of the patients enrolled in the BC early detection cohort were available for comparison and were examined with the same three assays used to probe the urinary cell samples. In 91 (89%) of these 102 cancers, at least one mutation in the eleven genes studied was present (in the 10-gene panel or the *TERT* promoter). Moreover, at least one of the mutations identified in the urine samples from these 102 patients was also identified in 83% of the corresponding primary BC samples (*Supplementary files 5* and *6*).

Analysis of these tumors also shed light on the basis for 'false negatives,' the urine samples with no detectable mutations from patients who ultimately developed BC. We attributed false negatives to the possibility that the corresponding BC either did not harbor a mutation in any of these 11 genes or the fraction of neoplastic cells in the urine sample was insufficient to allow detection with the assays used. We identified a mutation in at least one of the 11 genes in 62% of the primary tumors from patients with false negative urine tests (*Supplementary file 3* and *8*). We concluded that 62% of the 29 false negative tests were due to insufficient cancer cells in the urine while the remaining 38% were due to the absence of any of the queried mutations in the primary tumor tissue.

## UroSEEK: biomarkers in combination

The ten-gene multiplex assay, the *TERT* singleplex assay, and the aneuploidy assays yielded 68%, 57%, and 46% sensitivities, respectively, when used separately (*Table 1a* and *Supplementary files 5*, *6*, and *7*). Sensitivity was increased when the three assays were performed on each urine cell

sample. In samples without *TERT* promoter mutations (*n* = 45), mutations in one of the other ten genes were detected (*Figure 4* and *Supplementary file 5*). Conversely, 35 samples negative for mutations in the multiplex assay were detected by virtue of *TERT* promoter mutations (*Figure 4* and *Supplementary file 6*). Finally, ten of the urinary cell samples without any detectable mutations in the 11 genes were positive for aneuploidy (*Figure 4* and *Supplementary file 7*). Thus, when the three assays were used together (test termed 'UroSEEK'), and a positive result in any one of the assays was sufficient to score a sample as positive, the sensitivity rose to 83% (95% CI, 76% to 88%). Only one of the 188 samples from healthy individuals was scored positive by UroSEEK (specificity 99.5%, CI 97% to 100%). Twenty-six (6.5%) of the 395 patients in the BC early detection cohort who did not develop BC during the course of the study scored positive by the UroSEEK test (specificity 93%, CI 91% to 96%). On average, UroSEEK positivity preceded the diagnosis of BC by 2.3 months, and in eight cases, by >one year (*Figure 5* and *Supplementary file 2*).

## UroSEEK plus cytology

As both cytology and UroSEEK are non-invasive tests and can be performed on the same urine sample, we assessed their performance in combination. Cytology was available for 347 patients in the BC early detection cohort (*Table 1a* and *Supplementary file 2*). Among the 40 patients who developed biopsy-proven cancer in this cohort, cytology was positive in 17 cases (43% sensitivity), and UroSEEK was positive in 100% of these cancer patients. UroSEEK was also positive in 95% of 23 cancer patients whose urines were negative by cytology. Thus, in combination, UroSEEK plus cytology achieved 95% (95% CI, 83% to 99%) sensitivity, a 12% increase over UroSEEK and a 52% increase over cytology. Finally, none of the 299 patients who did not develop cancer over the course of the study were positive by cytology (100% specificity), but 20 (6.6%) were positive by UroSEEK, giving the combination of UroSEEK and cytology a specificity of 93% (95% CI, 90% to 96%).

## UTUC cohort

### Cohort characteristics

The gender distribution of this cohort, 32 females and 24 males, is atypical of UTUC patients in Western countries where males predominate (*Shariat et al., 2011*), but is consistent with earlier epidemiologic studies of Taiwanese individuals with known exposures to AA (summary in *Table 2*; individual data in *Supplementary file 9*)(*Chen et al., 2012*). Tobacco use was reported by 18% in this cohort and were all males. Based on estimated glomerular filtration rate (eGFR) values, renal function was unimpaired (chronic kidney disease (CKD) stage 0–2) in 45% of the subjects, while mild-to-moderate renal disease (CKD stage 3) or severe disease (CKD stages 4–5) was noted for 43% and 12% of the cohort, respectively (*Table 2*). Tumors were confined to a single site along the upper urinary tract in the majority of cases (38% renal pelvis; 39% ureter), while multifocal tumors affecting

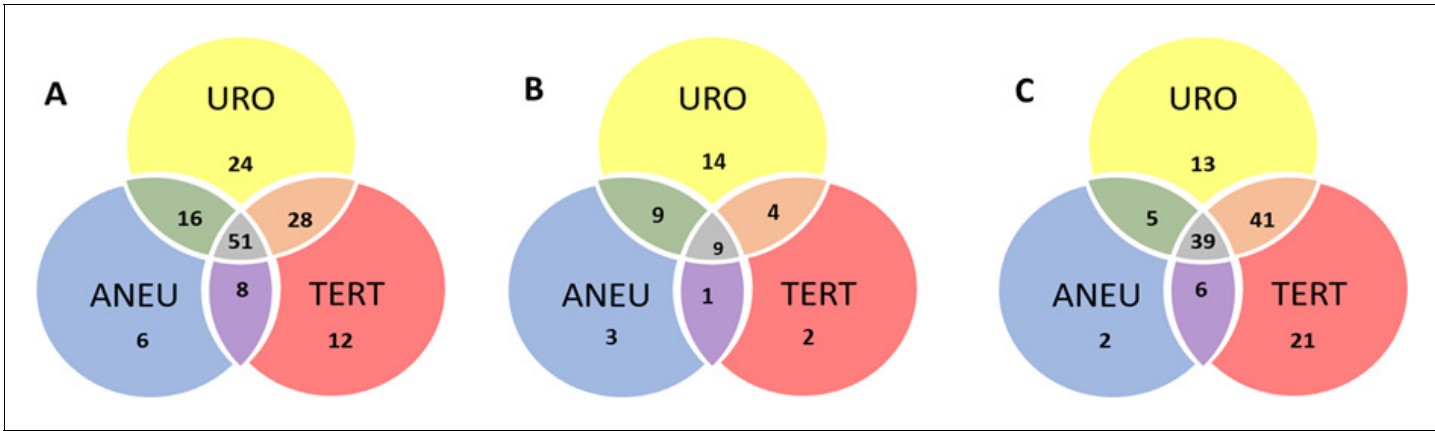

**Figure 4.** Venn diagram showing the distribution of positive results for each of the three UroSEEK assays for the (A) BC early detection (B) UTUC and (C) BC surveillance cohorts.  URO = Ten gene panel, TERT = *TERT* promoter region, ANEU = Aneuploidy test.
DOI: https://doi.org/10.7554/eLife.32143.006

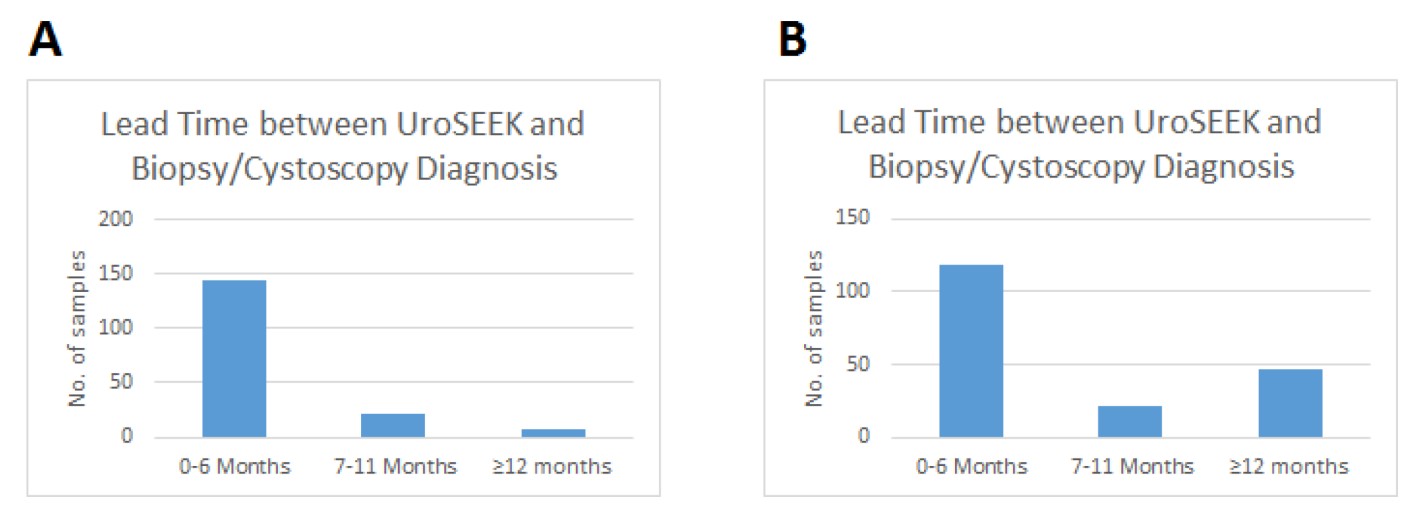

**Figure 5.** Bar graphs of the lead time between a positive UroSEEK test and the detection of disease at the clinical level in the (**A**) BC early detection and (**B**) BC surveillance cohorts.
DOI: https://doi.org/10.7554/eLife.32143.007

both renal pelvis and ureter occurred in 23% of the patients. Synchronous bladder cancer (diagnosed within 3 months prior to nephroureterectomy) was present in 38% of patients. Tumors were classified as high grade in 89% of the cases, with the majority categorized as muscle-invasive (T2-T4, 66%; *Table 2*).

## Genetic analysis

The multiplex assay detected at least one mutation in 36 of the 56 urinary cell samples from UTUC patients (64%, 95% CI, 51% to 76%; *Table 2* and *Supplementary file 10*). A total of 57 mutations were detected in nine of the ten target genes (*Figure 3*). The median MAF in the urinary cells was 5.6% and ranged from 0.3% to 80%. The most commonly altered genes were *TP53* (n = 33, 58% of the 57 mutations) and *FGFR3* (n = 9, 16% of the 57 mutations; *Figure 3*).

Mutations in the *TERT* promoter were detected in 16 of the 56 urinary cell samples from UTUC patients (29%, 95% CI, 18% to 42%; *Table 2* and *Supplementary file 11*). The median *TERT* MAF in the urinary cells was 2.22% and ranged from 0.59% to 46.3%. One of the 188 urinary samples from healthy individuals harbored a mutation (*TERT* g.1295250C > T with a MAF of 0.39%). In the UTUC urinary cell samples, most of the TERT mutations (94%) were at one of two positions, TERT: g.1295228 (67%) and TERT:g.1295250 (28%), which are 66 and 88 bp upstream of the transcription start site, respectively. A third position, TERT:g.1295242, was also involved in the remaining 6% of cases.

Aneuploidy was detected in 22 of the 56 urinary cell samples from UTUC patients (39%, 95% CI, 28% to 52%, *Supplementary file 12* and *13*). The most commonly altered chromosome arms were 1q, 7q, 8q, 17 p, and 18q.

## Comparison with primary tumors

The distribution of mutant genes in primary tumors (*Supplementary file 14*) was consistent with findings from some (*Hoang et al., 2013*; *Lee et al., 2018*; *Yuan et al., 2016*) but not all (*Moss et al., 2017*; *Sfakianos et al., 2015*), exome-wide and targeted sequencing studies of UTUCs. In the present study, *TP53* mutations were found only in high-grade UTUCs, while *FGFR3* mutations dominated in low-grade tumors (present in 5/6). Such mutational patterns have been previously reported by others (*Sfakianos et al., 2015*). However, the overall frequency of *FGFR3* mutations in our UTUC cohort (21%) was relatively low compared to values reported by *Moss et al. (2017)* (74%) and *Sfakianos et al. (2015)* (54%), but was comparable to values reported by *Hoang et al., 2013* (8%) and *Lee et al. (2018)* (13%). We attribute this difference to the race/ethnicity profile of the

**Table 2.** Demographic, clinical and genetic features of the UTUC cohort stratified by UroSEEK results.

| | N | % | Ten-gene multiplex positive | TERT positive | Aneuploidy positive | UroSEEK positive |
|---|---|---|---|---|---|---|
| All subjects | 56 | 100% | 64% | 29% | 39% | 75% |
| **Gender** | | | | | | |
| Males | 24 | 43% | 71% | 33% | 54% | 83% |
| Females | 32 | 57% | 59% | 25% | 28% | 69% |
| **CKD stage** | | | | | | |
| 0–2 | 25 | 45% | 68% | 36% | 44% | 76% |
| 3A | 14 | 25% | 50% | 21% | 43% | 71% |
| 3B | 10 | 18% | 80% | 20% | 40% | 80% |
| 4 | 4 | 7% | 25% | 50% | 0% | 50% |
| 5 | 3 | 5% | 100% | 0% | 33% | 100% |
| **Tumor grade** | | | | | | |
| Low | 6 | 11% | 67% | 50% | 17% | 67% |
| High | 50 | 89% | 64% | 26% | 42% | 76% |
| **Tumor stage** | | | | | | |
| Ta | 11 | 20% | 73% | 55% | 45% | 82% |
| T1 | 8 | 14% | 50% | 0% | 38% | 75% |
| T2 | 10 | 18% | 80% | 20% | 10% | 80% |
| T3 | 24 | 43% | 67% | 33% | 54% | 79% |
| T4 | 3 | 5% | 0% | 0% | 0% | 0% |
| **Upper urinary tract tumor site** | | | | | | |
| Lower ureter | 17 | 30% | 76% | 18% | 35% | 76% |
| Upper ureter | 1 | 2% | 100% | 0% | 0% | 100% |
| Ureterovesical junction | 2 | 4% | 0% | 0% | 0% | 0% |
| Lower ureter and upper ureter | 2 | 4% | 100% | 50% | 50% | 100% |
| Renal pelvis | 21 | 38% | 57% | 38% | 38% | 76% |
| Renal pelvis and lower ureter | 4 | 7% | 75% | 25% | 50% | 100% |
| Renal pelvis and upper ureter | 5 | 9% | 40% | 40% | 60% | 60% |
| Renal pelvis, lower ureter, upper ureter | 4 | 7% | 75% | 25% | 50% | 75% |
| **Synchronous bladder cancer** | | | | | | |
| Present | 21 | 38% | 52% | 29% | 33% | 62% |
| Absent | 35 | 63% | 71% | 29% | 43% | 83% |
| **UTUC risk factors** | | | | | | |
| Aristolactam-DNA adducts present | 54 | 96% | 65% | 30% | 39% | 74% |
| Smoking history | 10 | 18% | 70% | 30% | 60% | 70% |

CKD, chronic kidney disease.

DOI: https://doi.org/10.7554/eLife.32143.008

cohorts under comparison, as *FGFR3* mutation levels are relatively low in UTUCs from Han Chinese patients (3–9%) compared to Western patients (36–60%), as reported by *Yuan et al. (2016)*. Our cohort was Taiwanese and principally of Han Chinese descent, as were the *Hoang et al., 2013* (Taiwanese) and *Lee et al. (2018)* (Korean) cohorts, whereas Western patients were examined in the *Sfakianos et al. (2015)* and *Moss et al. (2017)* studies.

Tumor samples from all 56 patients with UTUC were available for comparison and were subjected to the same three assays used to analyze the urinary cell samples. At least one mutation could be identified in the urinary cells from 39 UTUC cases. In 35 (90%) of these 39 cases, at least one of the mutations identified in the urine sample (*Supplementary file 10* and *11*) was also identified in the

corresponding tumor DNA sample (*Supplementary file 14* and *15*). When all 80 mutations identified in the urinary cells were considered, 63 (79%) were identified in the corresponding tumor sample. The discrepancies between urine and tumor samples in any of the three assays might be explained by the fact that we had access to only one tumor per patient, even though more than one anatomically distinct tumor was often evident clinically (*Table 2*). In addition, DNA was extracted from only one location in each tumor; thus, intratumoral heterogeneity (*McGranahan et al., 2015*) could have been responsible for some of the discrepancies.

The tumor data also helped to establish why 17 of the 56 urinary cell samples from UTUC patients did not contain detectable mutations. From the evaluation of the primary tumor samples, we found that only four (24%) of these 17 urine samples were from patients whose tumors did not contain any of the queried mutations (*Supplementary file 14* and *15*). Thus, we concluded that the main reason for failure of the mutation test was an insufficient number of cancer cells in the urine, which accounted for 13 (76%) of the 17 failures.

Aneuploidy was observed in 22 of the urinary cell samples (*Supplementary file 12*). Overall, 96% of the chromosomal gains or losses observed in the urinary cells were also observed in the primary tumors (*Supplementary file 13*). Conversely, aneuploidy was not observed in 34 of the urinary cell samples. Evaluation of the corresponding 56 tumors with the same assay demonstrated that all but three were aneuploid. Therefore, as with mutations, the main reason for failure of the aneuploidy assay was insufficient neoplastic DNA in the urinary cells.

## UroSEEK: biomarkers in combination

The ten-gene multiplex assay, the *TERT* singleplex assay, and the aneuploidy assays yielded 64%, 29%, and 39% sensitivities, respectively, when used separately in the UTUC cohort (*Table 2*). Mutations in one of the other ten genes were detected in 23 samples without TERT promoter mutations (*Figure 4*). Conversely, three samples without detectable mutations in the multiplex assay scored positive for *TERT* promoter mutations (*Figure 4*). Furthermore, three of the urinary cell samples without any detectable mutations were positive for aneuploidy (*Figure 4*). Thus, when the three assays were used together, and a positive result in any one assay was sufficient to score a sample as positive, the sensitivity rose to 75% (95% CI 62.2% to 84.6%).

To determine the basis for the increased sensitivity afforded by the combination assays, we evaluated data from the primary tumors of the three patients whose urinary cell samples exhibited aneuploidy but did not harbor detectable mutations. We found that these three tumors did not contain any mutations in the 11 queried genes, which explained why these same assays were negative when applied to the urinary cell DNAs. These three tumors were aneuploid, thus enabling their detection through copy number variations in the urinary cell samples.

## Correlation with clinical features

The most clinically desirable biomarkers are those associated with early stage tumor development as they enable surgical removal of lesions before widespread metastasis. In UTUC, ten-year cancer specific survival rates show that 91% of patients with stage T1 malignancies are expected to be cured by surgery, compared to only 78%, 34% and 0% of patients with stage T2, T3, or T4 tumors, respectively (*Li et al., 2008*). In our cohort, UroSEEK was equally sensitive for detecting early and late UTUCs. The test was positive in 15 (79%) of 19 patients with stage Ta/T1 tumors and 27 (73%) of 37 patients with stage T2-T4 tumors (*Table 2*). Sensitivity was comparable across gender, tumor grade, tumor location and risk factors for developing UTUC (*Table 2*), indicating that the assay was suitable for evaluation of diverse patient populations. UroSEEK performance was also comparable in UTUC cohorts with and without synchronous BC (*Table 2*).

UroSEEK was also considerably more sensitive than urine cytology in the UTUC cohort (*Figures 2* and *6*). Cytology was available in 42 cases, and of these, four (9.5%) were positive on cytology. UroSEEK detected all four of these cases. In addition, UroSEEK was positive in 5/7 cases that had an equivocal cytology diagnosis of suspicious for malignancy, and 22/31 samples that were negative on cytology.

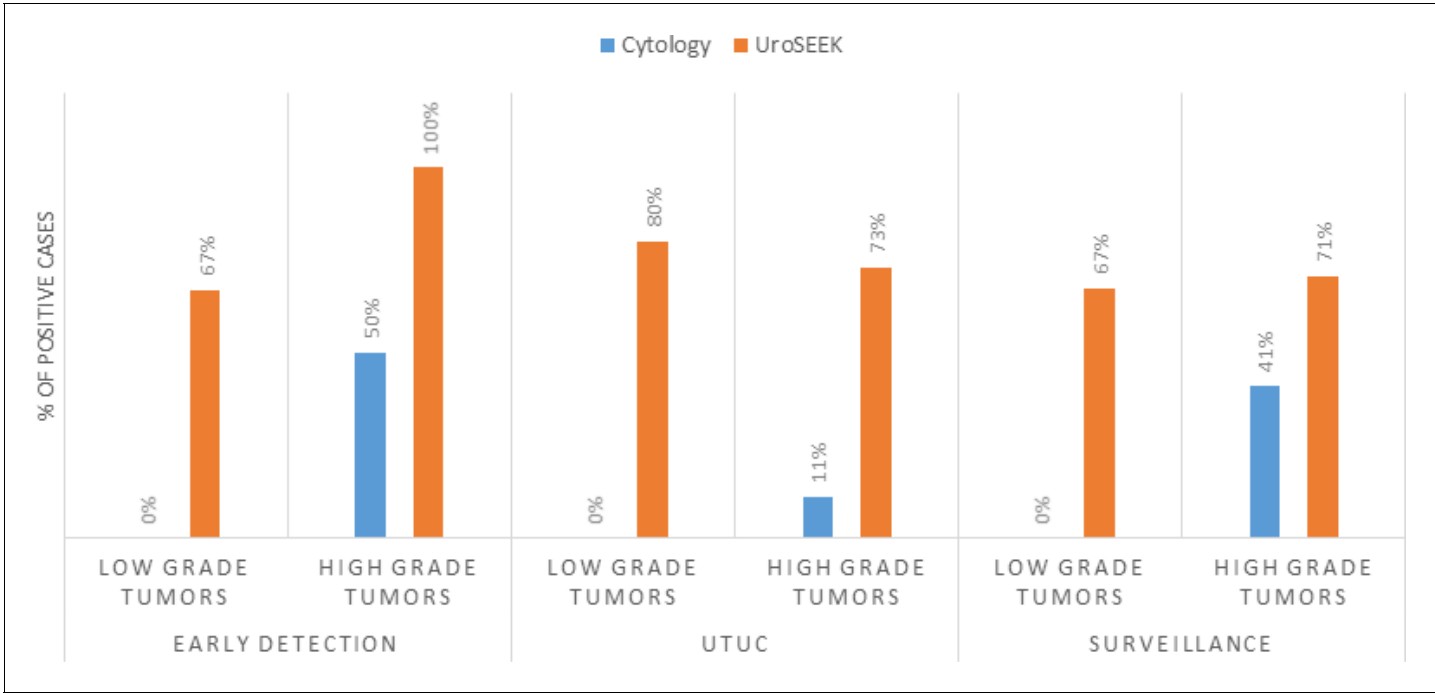

**Figure 6.** Bar graphs representing the performance of Cytology vs. UroSEEK in diagnosis of low- and high-grade urothelial neoplasms in the early detection and surveillance BC cohorts and the UTUC cohort.

DOI: https://doi.org/10.7554/eLife.32143.009

## Aristolochic acid exposure

The activated metabolites of AA bind covalently to the exocyclic amino groups in purine bases, with a preference for dA, leading to characteristic A > T transversions (*Hollstein et al., 2013*; *Moriya et al., 2011*). To determine whether individuals in the UTUC cohort had been exposed to AA, we quantified renal cortical DNA adducts using mass spectrometry (*Yun et al., 2012*). All but two of the 56 patients had detectable aristolactam (AL)-DNA adducts (*Table 2*) with levels ranging from 0.4 to 68 dA-AL-I adducts per $10^8$ nucleotides. Moreover, the A > T signature mutation (*Hoang et al., 2013*) associated with AA was highly represented in the mutational spectra of *TP53* (18/32 A > T) and *HRAS* (2/2 A > T) found in urinary cells (*Supplementary file 10*).

## BC surveillance cohort

### Cohort characteristics

Our strategy for BC surveillance was different than for early detection of BC. In these patients, a BC was surgically excised for treatment and diagnosis. Tumor tissue was thus generally available, and in most such tumors, a mutation was identified. For example, we found during the course of this study that a mutation in at least one of the 11 queried genes was present in 95.2% of BCs evaluated. We evaluated a total of 322 patients with a BC tumor containing a mutation in at least one of the 11 genes and a urine sample collected within 0–5 years after surgery. We determined whether a single urine sample taken a relatively short time following surgical excision of the BC could reveal residual disease in these 322 patients, as evidenced by later recurrence. In 187 (58%) of the 322 patients, clinically evident BC developed after a median follow-up period of 10.7 months (range 0 to 51 months). The histopathologic types and tumor stages of these patients are summarized in *Table 1b* and detailed in *Supplementary file 16*. The median age of the participants was 62 years (range 20 to 93), and 75% of the patients were male as expected from the demographics of BC.

### Genetic analysis

The multiplex assay detected mutations in 52% of the urinary cell samples from patients who developed recurrent BC during the study interval (95% CI, 45% to 60%; *Supplementary file 16* and

*Supplementary file 17*). The median MAF in the urinary cells with detectable mutations was 7% (6.89%). The most commonly altered genes were *FGFR3* (43% of the 134 mutations) and *TP53* (30% of the 134 mutations; *Figure 3*). Some cases were however considered to be false positives; 7% of the 135 patients who did not develop recurrent BC during the course of the study had a detectable mutation in their urinary cell sample (see Discussion). The mean interval between a positive multiplex assay test and the diagnosis of recurrent BC was 7 months (range 0 to 51 months).

Mutations in the *TERT* promoter were detected in 57% of the urinary cell samples from patients who developed recurrent BC during the study interval (95% CI 44% to 58%; (*Table 1b* and *Supplementary file 18*). The median *TERT* MAF in the urinary cells with detectable mutations was 5% (5.02%). Mutations were detected in the same three promoter positions observed in the urinary cells of the early detection cohort. The mean interval between a positive *TERT* test and the diagnosis of recurrent BC was 7 months (range 0 to 40 months). Some results were considered to be false positives; 10% (95% CI, 83% to 94%) of the 135 patients who did not develop recurrent BC during the course of the study had a detectable *TERT* promoter mutation in their urine sample.

Aneuploidy was detected in 28% (95% CI 24% to 37%) of the urinary cell samples from the patients who developed recurrent BC during the course of the study (*Table 1b* and *Supplementary file 19*). The most commonly altered chromosome arms were 8 p, 8q, and 9 p, as in the early detection cohort. In this assay, only 2% of the 135 patients who did not develop recurrent BC during the course of the study exhibited aneuploidy in at least one of their urinary cell samples.

## UroSEEK: biomarkers in combination

The ten-gene multiplex assay, the *TERT* singleplex assay, and the aneuploidy assays yielded 52%, 57%, and 28% sensitivities, respectively, when used separately on the BC surveillance cohort (*Table 1b* and *Supplementary file 17*, *18* and *19*). Thirty-two samples without *TERT* promoter mutations were detected by mutations in one of the other ten genes (*Figure 4* and *Supplementary file 17*). Conversely, 41 samples without detectable mutations in the multiplex assay had *TERT* promoter mutations. Finally, aneuploidy was detected in three of the urinary cell samples without mutations in any of the 11 genes. Thus, the sensitivity of UroSEEK was 68% (95% CI, 59% to 73%; *Table 1b*). Twenty percent of the 135 patients in this cohort who did not develop BC during the course of the study scored positive by the UroSEEK test, yielding a specificity of 80% (95% CI, 77% to 91%). On average, UroSEEK positivity preceded the diagnosis of BC by 7 months, and in 47 cases, by >one year (*Figure 4* and *Supplementary file 16*).

Cytology was available for 196 patients in the BC surveillance cohort (*Supplementary file 16*). Among the 120 patients who developed recurrent BC in this cohort, 30 (25%) were positive by cytology. Conversely, no positive cytology results were observed in patients without recurrent tumors. UroSEEK was positive in 90% of the recurrent BC patients with urines positive by cytology and in 61% of the 90 recurrent BC patients with urines negative by cytology. Thus, in combination, UroSEEK plus cytology achieved 71% sensitivity (95% CI, 61.84% to 78.77; *Figure 2* and *Supplementary files 17*, *18*, *19*). Among the 76 patients with cytology who did not develop recurrent BC during the course of the study, 18% scored as positive by either cytology or UroSEEK, which yielded a specificity of 82% (95% CI, 71% to 90%; see Discussion).

## Low- vs. high-grade urothelial neoplasms (both BC cohorts)

The advantage of UroSEEK over cytology was particularly evident in low-grade BC (Papillary urothelial neoplasms of low malignant potential and non-invasive low-grade papillary urothelial carcinomas). Cytology was available for 49 low-grade tumors evaluated in this study (six from the early detection cohort and 43 from the Surveillance cohort). None of these low-grade tumors, however, were detected by cytology (0% sensitivity; 95% CI, 0.0% to 6.7%). In contrast, UroSEEK detected 67% (95% CI 51% to 81%) of the low-grade tumors (identical rate of 67% in both cohorts; *Supplementary file 20* and *Figure 6*). Cytology was also available for 102 high-grade tumors (in situ urothelial carcinoma, non-invasive high-grade papillary urothelial carcinoma or infiltrating high-grade urothelial carcinoma) evaluated in this study (early detection cohort, *n* = 34, and BC Surveillance cohort, *n* = 68). Cytology was positive in 45% of these patients (50% and 41% in the early detection and BC surveillance cohorts, respectively) while UroSEEK was positive in 80% (100% and 71% in the early detection and surveillance cohorts, respectively; *Supplementary file 2* and *16*).

## Discussion

Cytology is a non-invasive test that is highly specific, and in expert hands nearly always indicates the presence of urothelial malignancy when positive. This specificity was verified in our study: all 51 patients in the BC early detection cohort whose urine samples were positive by cytology developed biopsy-proven BC. However, cytology is not particularly sensitive. UroSEEK adds considerably to sensitivity, as it raised the sensitivity of cytology from 43% to 95% in the BC early detection cohort, from 25% to 71% in the BC Surveillance cohort, and from 10% to 75% in the UTUC cohort. The increased sensitivity was further highlighted by the fact that UroSEEK-positive results preceded clinical diagnosis or positive cytology by months to years in the BC surveillance cohort.

The advantage of using UroSEEK in addition to cytology was particularly evident for low-grade tumors. Cytology was negative in all 49 cases in the BC early detection cohort, while 2/3 of these patients were positive with UroSEEK. Similarly, UroSEEK correctly identified 80% of low-grade UTUC while none were detected by cytology. Another example of the utility of the combination of Uro-SEEK plus cytology was evident in patients with an equivocal cytology reading. A relatively large number of urine samples receive such an equivocal cytologic reading, even in the hands of a sub-specialized, board-certified cytopathology expert such as employed in the current study (*Barkan et al., 2016*). In the BC early detection cohort, for example, 105 urine samples were scored as 'atypical', and of these cases, 19% developed recurrence while the other 81% did not. UroSEEK was positive in 95% of the atypical cases that developed BC, but only in 13% of the atypical cases that did not develop cancer. These results demonstrate that UroSEEK can be used to more confidently interpret atypical cytology results.

Although UroSEEK is more sensitive than cytology, it is less specific. In this study, we assessed specificity in several independent ways. The first, and in some ways, most straightforward, was in a collection of urine samples from healthy individuals. In these 188 individuals, only one sample was positive, yielding a specificity of 99.5% (CI 97% to 100%). Such high specificity can be considered the technical specificity of the test, but biological specificities are also important. In the BC early detection cohort, 26 of the 395 patients who did not develop BC scored positive, yielding a specificity of 93% (CI 90.50% to 96%), or a false positive rate of 6.5%. These 'false positives' detected by UroSEEK could result from several factors. First, we cannot be certain that the patients whose urinary cells harbored genetic alterations did not have cancer. The follow-up period for many of patients was only one year, and cystoscopy was not generally performed. Second, it is possible that there are clonal proliferations in the bladder epithelium that increase with age. The patients in the BC early detection cohort were on average older than the 188 healthy individuals used as controls (40 years vs. 58 years). Although this explanation is speculative, clonal proliferations that are not considered neoplastic have been described in the bone, skin, and other tissues (*Risques and Kennedy, 2018*). Clonal proliferations may also be the basis for any discordance between mutations in urinary cells and in the primary tumors of the same patients. Although in the majority of cases, at least one of the mutations identified in the urine was also present in the primary tumor, this was not true in 22% of the cases in the BC early detection cohort. In these cases, UroSEEK could be detecting clonal proliferations in the bladder epithelium that did not progress to cancer, and such proliferations may be more common in patients with BC than in the general population *Risques and Kennedy, 2018*Because only one biopsy from the primary tumor was available for comparison, it is also possible that intratumoral heterogeneity explains some of the discrepancies. False positives in the BC surveillance cohort could be explained in similar ways. False positives are not unique to our study; they have been observed in all other molecular assays for BC, including FDA-approved tests (*Dimashkieh et al., 2013*; *Gopalakrishna et al., 2017*; *Hajdinjak, 2008*). Whether the false positives in these other assays have the same biological basis is an important area for future research.

There are two factors that limit sensitivity for genetically-based biomarkers. First, a sample can only be scored as positive for the biomarker if it contains DNA from a sufficient number of neoplastic cells to be detected by the assay. Second, the tumor from which the neoplastic cells were derived must harbor the genetic alteration that is queried. Combination assays can increase sensitivity by assessing more genetic alterations, and are thereby more likely to detect at least one genetic alteration present in the tumor. However, mutations in clinical samples are often present at low allele frequencies (*Supplementary files 5*, *6*, *10* and *11*), requiring high coverage of every base queried. It would be prohibitively expensive to perform whole exome sequencing at 10,000x coverage, for

example, so some compromise is needed. In our study, we evaluated carefully selected regions of 11 genes, including TERT, together with copy number analysis of 39 chromosome arms. Even if a tumor does not contain a genetic alteration in one of the 11 genes assessed, it might still be detectable by the urinary cell assay for aneuploidy. The sensitivity of aneuploidy detection however is less than that of the mutation assays. Simulations demonstrated that DNA containing a minimum of 1% neoplastic cells is required for reliable aneuploidy detection, while mutations present in as few as 0.03% of the DNA templates can be detected by the mutation assays used in our study (*Bettegowda et al., 2014*; *Kinde et al., 2013*; *Wang et al., 2016*). Nevertheless, urinary cell samples that had relatively high fractions of neoplastic cells but did not contain a detectable mutation in the 11 queried genes should still be detectable due to their aneuploidy. In addition, some of the mutations in the 11 genes queried, such as large insertions or deletions or complex changes, might be undetectable by mutation-based assays, but such samples might still score positive in a test for aneuploidy.

For UTUC, although the approach described here has significant potential for screening purposes, we emphasize that the current study demonstrates proof-of-principle rather than clinical applicability given the small number of patients evaluated. Another caveat is that our assays cannot distinguish between UTUC and BC. We consider this a strength of the assay since the detection of BC is equally important given that patients exposed to AA are at risk for BC as well as for UTUC (*Poon et al., 2015*). It has been estimated that 100 million people in China are at risk for UTUC as a result of exposure to this carcinogen (*Grollman, 2013*; *Hu et al., 2004*). Non-invasive, sensitive methods to screen the large numbers of at-risk individuals for UTUC in such populations are thus clearly desirable. UroSEEK could also be used to monitor for UTUC recurrence in bladder, which occurs in 22% to 47% of cases, or in the contralateral tract affecting 2% to 6% of patients (*Rouprêt et al., 2015*) and up to 30% of AA-related UTUC patients (*Chen et al., 2013*). Currently, no such screening methods are available, as illustrated in the current study where urine cytology failed to detect 90% of UTUC cases. Radiologic tests, such as MRI or CT-scans, are not well suited for screening, and the latter confers significant radiation exposure. Ureteroscopy is often definitive, but in addition to being invasive, requires highly skilled clinicians and is also ill-suited as a screening tool (*Golan et al., 2015*).

Liquid biopsy has recently gained attention as a non-invasive approach to screen for cancer. Although this concept often refers to blood samples, it can be applied to other body fluids, such as urine (*Patel et al., 2017*; *Sidransky et al., 1991*; *Togneri et al., 2016*). Urine contains DNA from several sources, including (i) glomerular filtration of circulating free DNA (*Botezatu et al., 2000*) released by normal and tumor cells from sites throughout the body; (ii) DNA released directly into urine by normal and tumor cells of the urinary tract; and (iii) intact normal or malignant cells of the urinary tract exfoliated into urine (*Bettegowda et al., 2014*; *Dawson et al., 2013*; *Dressman et al., 2003*; *Forshew et al., 2012*; *Haber and Velculescu, 2014*; *Kinde et al., 2013*; *Springer et al., 2015*; *Vogelstein and Kinzler, 1999*; *Wang et al., 2015a*; *Wang et al., 2015b*; *Wang et al., 2016*). We chose the latter option for the current study to increase sensitivity and specificity.

While optimizing conditions for the current study, we compared the relative performance of mutation assays in matched plasma and urine samples obtained from 14 UTUC patients. In each case, a *TERT* or *TP53* mutation was first identified in the primary tumor. That particular mutation was subsequently queried in DNA from the urine and plasma using a singleplex assay. Mutations were detected in 93% (13/14) of the urinary cell DNA samples compared to 36% (5/14) of the plasma samples. Importantly, the plasma test failed to identify any of the six non-muscle-invasive cancers (Ta/T1), while all six (100%) were identified in the matched urinary cell DNA samples. The superior performance in urinary cells was likely due to a substantial enrichment for mutated DNA in these cells compared to plasma; the median MAF in plasma when a mutation was detectable was only 0.3% compared to 15% in the urinary cells.

Our study lays the conceptual and practical framework for a novel test that could be used in the management of patients with urothelial cancer. Large prospective trials will be required to demonstrate the ability of UroSEEK to improve the management of patients with hematuria or dysuria or patients at risk for urothelial cancer recurrence. Before carrying out large-scale trials to evaluate such clinical utility, it is informative to predict what the performance characteristics of such a test might be. As one example, consider the use of UroSEEK plus cytology in patients presenting to their physician with microscopic hematuria or dysuria, a commonly encountered situation. In large population-

based studies involving over 80,000 individuals participating in health screening, the fraction of individuals with micro-hematuria ranged from 2.4% to 31.1% (*Davis et al., 2012*; *Wein et al., 2012*). It has been estimated that 5% of such patients actually have urothelial cancer (*Khadra et al., 2000*). In the current study, UroSEEK plus cytology had a sensitivity of 95% and a specificity of 93% for BC in patients with this presentation. These results extrapolate to a positive predictive value (PPV) of 66% (95% CI, 55% to 74%) and a negative predictive value (NPV) of 99.3% (95% CI, 97.3% to 99.8%). These values are well above those generally considered to be diagnostically helpful and are considerably higher than achieved in FDA-approved tests for this indication (*Dimashkieh et al., 2013*; *Hajdinjak, 2008*).

We envision that the first application of UroSEEK would be patients such as those described here in the cohorts used for early detection of BC and UTUC. Patients with hematuria who might otherwise be referred to cystoscopy would be tested by UroSEEK plus cytology. Such tests could be ordered by general practitioners and do not require consultation with a urologist. Only if a test was positive would cystoscopy be required. The sensitivities, specificities, and PPV and NPV of UroSEEK plus cytology suggest that this strategy is well within the boundaries of currently accepted medical guidelines. Optimistically, 95% of patients would be spared the discomfort and inconvenience of cystoscopy as well as its unintended consequences. Only patients who have positive UroSEEK, persistent symptoms, or hematuria would undergo cystoscopy. The savings in this approach would be considerable, as we estimate the cost of UroSEEK plus cytology to be less than 1/3 of the cost of cystoscopy.

## Materials and methods

### Patients and samples

#### BC early detection and BC surveillance cohorts

Urine samples were collected prospectively from patients in four participating institutions, including Johns Hopkins Hospital, Baltimore, MD, USA; A.C. Camargo Cancer Center, Sao Paulo, Brazil; Osaka University Hospital, Osaka, Japan; and Hacettepe University Hospital, Ankara, Turkey. The study was approved by the Institutional Review Boards of Johns Hopkins Hospital and all other participating institutions. Material transfer agreements were obtained. Patients with a known history of malignancy other than BC were excluded from the study. The BC early detection cohort comprised patients who were referred to a urology clinic in one of the above hospitals because of hematuria or lower urinary tract symptoms (570 patients; *Supplementary file 2*). The other cohort (322 patients) represented patients with prior established diagnosis of BC who are on surveillance for disease recurrence (BC surveillance cohort). As noted in the main text, the primary tumors from these patients harbored mutations in at least one of the 11 genes assessed through the multiplex or singleplex assays. A minimum of 12 months of follow-up from the date of urine collection was required in cases with no evidence of incident (BC early detection cohort) or recurrent tumors (BC Surveillance cohort) to be included in the study. Urine samples were collected before any procedures, such as cystoscopy, were performed during patient visits. A total of 892 urine samples were analyzed and composed of two types of samples. The first was residual urinary cells after processing with standard BD SurePath liquid-based cytology protocols (Becton Dickinson and Company; Franklin Lakes, NJ, USA). To allow for standard-of-care, residual SurePath fluids were kept refrigerated for 6–8 weeks before submission for DNA purification to allow for any potential need for repeat cytology processing of the same sample. The second sample type was composed of bio-banked fresh urine samples in which 15–25 mL of voided urine samples were stored at 4°C for up to 60 min prior to centrifugation (10 min at 500 *g*) and the pellets stored at minus 80°C before DNA purification.

Formalin-fixed, paraffin-embedded (FFPE) tumor tissue samples from trans-urethral resections (TURB) or cystectomies were collected in 413 of the 892 cases. When several different tumors from the same patient were available (because of recurrences), the earliest tumor tissue obtained following donation of the urine sample was used in the early detection cohort. In the Surveillance cohort, tumors obtained before the donation of the urine sample were used in 146 of the 322 patients. In the other 176 Surveillance cases, the earliest tissue obtained following the donation of the urine sample was used. A genitourinary pathologist reviewed all histologic slides to confirm the diagnosis and select a representative tumor area with as high tumor cellularity as possible for that case.

Corresponding FFPE blocks were cored with a sterile 16-gauge needle. One to three cores were obtained per tumor and placed in 1.5 mL sterile tubes for DNA purification, as previously described (*Kinde et al., 2013*). Electronic medical records were reviewed to obtain medical history and follow up data in all patients.

## UTUC cohort

Sequential patients with UTUC scheduled to undergo a radical unilateral nephroureterectomy at National Taiwan University Hospital, Taipei Taiwan, in 2012 - 2016 were asked to participate in the study. All patients provided informed consent using the consent form and study design reviewed and approved by the Institutional Review Boards at National Taiwan University and Stony Brook University. A total of 56 UTUC patients were enrolled in the study after excluding four patients with gross hematuria and one patient with a tumor-urine DNA mismatch by identity testing (see below).

UTUC patients provided urine samples (12 hr collection ($n = 10$); spot urines ($n = 41$); spot and 12 hr collection ($n = 4$); bladder wash ($n = 1$)) one day prior to surgery. Urinary cells were isolated by centrifugation at 581 $g$ for 10 min at room temperature, washed three times in saline using the same centrifugation conditions, and stored frozen until DNA was isolated using a Qiagen kit #937255 (Germantown, MD). DNA was purified from fresh-frozen resected samples of upper tract tumors and renal cortex by standard phenol-chloroform extraction procedures (*Chen et al., 2012*; *Jelaković et al., 2012*). One upper urinary tract tumor per patient was analyzed; for cases with tumors at multiple sites, renal pelvic tumors were preferentially selected whenever available. FFPE tumor samples were staged and graded by a urologic pathologist, and the presence of one or more UTUC was confirmed by histopathology for each enrolled individual. Pertinent clinical and demographic data were obtained by a chart review of each individual. eGFR was calculated with the MDRD equation (*Levey et al., 2006*) and used to determine CKD stage (*Levey et al., 2005*).

## DNA adduct analysis

AL-DNA adduct (7-(deoxyadenosin-$N^6$-yl) aristolactam I; dA-AL-I) levels in 2 µg of DNA from the normal renal cortex of UTUC patients were quantified with ultra-performance liquid chromatography–electrospray ionization/multistage mass spectrometry (UPLC-ESI/MS$^n$) with a linear quadrupole ion trap mass spectrometer (LTQ Velos Pro, Thermo Fisher Scientific, San Jose, CA) as described previously (*Yun et al., 2012*).

## Mutation analysis

Three separate assays were used to search for abnormalities in urinary cell DNA. First, a multiplex PCR was used to detect mutations in regions of ten genes commonly mutated in urologic malignancies: *CDKN2A, ERBB2, FGFR3, HRAS, KRAS, MET, MLL, PIK3CA, TP53, and VHL* (*Cancer Genome Atlas Research Network, 2014*; *Lin et al., 2010*; *Mo et al., 2007*; *Netto, 2011*; *Sarkis et al., 1995*; *Sarkis et al., 1994*; *Sarkis et al., 1993*; *Wu, 2005*). The 57 primer pairs used for this multiplex PCR were divided in a total of three multiplex reactions, each containing non-overlapping amplicons (*Supplementary file 4*). These primers were used to amplify DNA in 25 µL reactions as previously described (*Kinde et al., 2011*), except that 15 cycles were used for the initial amplification. Second, the *TERT* promoter region was evaluated. A single amplification primer was used to amplify a 73 bp segment containing the region of the *TERT* promoter known to harbor mutations in BC and UTUC (*Killela et al., 2013*; *Kinde et al., 2013*). The conditions used to amplify it were the same as those used in the multiplex reactions described above except that Phusion GC Buffer (Thermo-Fisher) rather than HF buffer was used, and 20 cycles were used for the initial amplification. The *TERT* promoter region was not included in the multiplex PCR because of its high GC content.

PCR products were purified with AMPure XP beads (Beckman Coulter, PA, USA) and 0.25% of the purified PCR products (multiplex) or 0.0125% of the PCR products (*TERT* singleplex) were then amplified in a second round of PCR, as described by *Wang et al., 2016*. PCR products from the second round of amplification were then purified with AMPure and sequenced on an Illumina instrument. For each mutation identified, the mutant allele frequency (MAF) was determined by dividing the number of uniquely identified reads with mutations (*Kinde et al., 2011*) by the total number of uniquely identified reads. Each DNA sample was assessed in two independent PCRs, for both the *TERT* promoter and multiplex assays, and samples were scored as positive only if both PCRs showed

the same mutation. The MAFs and number of UIDs listed in the Supplementary Tables refer to the average of the two independent assays. All coordinates are reported relative to genome reference hg19.

To evaluate the statistical significance of putative mutations, we assessed DNA from white blood cells (WBCs) of 188 unrelated healthy individuals. A variant observed in the samples from the BC or UTUC cohorts was only scored as a mutation if it was observed at a much higher MAF than observed in normal WBCs. Specifically, the classification of a sample's DNA status was based on two complementary criteria applied to each mutation: 1) the difference between the average MAF in the sample of interest and the corresponding maximum MAF observed for that same mutation in a set of controls; and 2) the Stouffer's Z-score obtained by comparing the MAF in the sample of interest to a distribution of normal controls. To calculate the Z-score, the MAF in the sample of interest was normalized based on the mutation-specific distributions of MAFs observed among all controls. Following this mutation-specific normalization, a P-value was obtained by comparing the MAF of each mutation in each well with a reference distribution of MAFs built from normal controls where all mutations were included. The Stouffer's Z-score was then calculated from the P-values of two wells, weighted by their number of UIDs. The sample was classified as positive if either the difference or the Stouffer's Z-score of its mutations was above the thresholds determined from the normal WBCs. The threshold for the difference parameter was defined by the highest MAF observed in any normal WBCs. The threshold for the Stouffer's Z-score was chosen to allow one false positive among the 188 normal urine samples studied (see below).

## Analysis of aneuploidy

Aneuploidy was assessed with FastSeqS, which uses a single primer pair to amplify ~38,000 loci scattered throughout the genome (*Kinde et al., 2012*). After massively parallel sequencing, gains or losses of each of the 39 chromosome arms covered by the assay were determined using a bespoke statistical learning method described in *Douville et al., 2018*. A support vector machine (SVM) was used to discriminate between aneuploid and euploid samples. The SVM was trained using 3150 low neoplastic cell fraction synthetic aneuploid samples and 677 euploid peripheral WBC samples. Samples were scored as positive when the genome-wide aneuploidy score was >0.7, and there was at least one gain or loss of a chromosome arm.

## Identity checks

A multiplex reaction containing 26 primers detecting 31 common SNPs on chromosomes 10 and 20 was performed using the amplification conditions described above for the multiplex PCR. The 26 primers used for this identity evaluation are listed in *Supplementary file 21*.

## Normal control samples

Urine samples from 188 healthy volunteers (19–60 years; mean age 26 years) were obtained and processed identically to the bio-banked fresh urine samples as described above. Urinary cell DNA from these 188 samples was used to assess the specificity of the UroSEEK test. WBC DNA from 94 normal individuals was used to evaluate the technical specificity of the PCR analysis.

## Statistical analysis

Performance characteristics of urine cytology, UroSEEK and its three components were calculated using MedCalc statistical software, online version (https://www.medcalc.org/calc/diagnostic_test.php). Confidence intervals (95%) were determined with an online GraphPad Software Inc. statistical calculator (https://www.graphpad.com/quickcalcs/confInterval1/) using the modified Wald method.

## Acknowledgements

This research was supported by grants from the National Science Council, Taiwan to CHC (104–2314-B-002–132) and to YSP (104–2314-B-002–121-MY3). We appreciate the clinical services provided by Dr. Kuo-How Huang, Shuo-Meng Wang, Huai-Ching Tai and Yuan-Ju Lee (Department of Urology, National Taiwan University Hospital). We are grateful for the generous support provided by Henry and Marsha Laufer, the Virginia and DK Ludwig Fund for Cancer Research, the

Commonwealth Foundation, the John Templeton Foundation, and the Conrad R Hilton Foundation. All sequencing was performed at the Sol Goldman Sequencing Facility at Johns Hopkins. This work was also supported by grants from the NIH (Grants CA-77598, CA 06973, GM 07309, and ES019564). The authors appreciate the medical illustrations skillfully designed by Kathleen Gebhart (Media Services, Stony Brook University). The authors appreciate the help in editing by Janice Nigro.

## Additional information

### Competing interests

Luis A Diaz: Member of the board of directors of Personal Genome Diagnostics (PGDx) and Jounce Therapeutics. LAD holds equity in PapGene, Personal Genome Diagnostics (PGDx) and Phoremost. He is a paid consultant for Merck, PGDx and Phoremost. LAD is an inventor of licensed intellectual property related to technology for ctDNA analyses and mismatch repair deficiency for diagnosis and therapy from Johns Hopkins University. These licenses and relationships are associated with equity or royalty payments to LAD. The terms of all these arrangements are being managed by Johns Hopkins and Memorial Sloan Kettering in accordance with their conflict of interest policies. In addition, in the past 5 years, LAD has participated as a paid consultant for one-time engagements with Caris, Lyndra, Genocea Biosciences, Illumina and Cell Design Labs. Nickolas Papadopoulos, Ken W Kinzler, Bert Vogelstein: Founder of Personal Genome Diagnostics and PapGene and advises Sysmex-Inostics. These companies and others have licensed technologies from Johns Hopkins, of which BV, KK, and NP are inventors on a patent (U.S. 20140227705 A1) and receive royalties. The terms of these arrangements are managed by the university in accordance with its conflict of interest policies. The other authors declare that no competing interests exist.

### Funding

| Funder | Grant reference number | Author |
| --- | --- | --- |
| National Science Council | 104-2314-B-002-132 | Chung-Hsin Chen<br>Yeong-Shiau Pu |
| National Science Council | 104-2314-B-002-121-MY3 | Chung-Hsin Chen<br>Yeong-Shiau Pu |
| National Institutes of Health | | Yuxuan Wang<br>Arthur P Grollman<br>Ken W Kinzler |
| Henry and Marsha Laufer | | Arthur P Grollman |
| John Templeton Foundation | | Cristian Tomasetti |
| Conrad N. Hilton Foundation | | Nickolas Papadopoulos |
| Ludwig Institute for Cancer Research | | Ken W Kinzler<br>Bert Vogelstein |
| Sol Goldman Pancreatic Research Foundation | | Bert Vogelstein |

The funders had no role in study design, data collection and interpretation, or the decision to submit the work for publication.

### Author contributions

Simeon U Springer, Resources, Formal analysis, Validation, Investigation, Writing—review and editing; Chung-Hsin Chen, Conceptualization, Investigation, Writing—review and editing, Project administration, Resources, Funding acquisition; Maria Del Carmen Rodriguez Pena, Resources, Formal analysis, Investigation, Writing—review and editing; Lu Li, Christopher Douville, Yuxuan Wang, Joshua David Cohen, Natalie Silliman, Joy Schaefer, Janine Ptak, Lisa Dobbyn, Maria Papoli, Robert J Turesky, Byeong Hwa Yun, Cristian Tomasetti, Formal analysis, Investigation; Diana Taheri, Aline C Tregnago, Resources, Formal analysis, Investigation; Isaac Kinde, Luis A Diaz, Conceptualization, Writing—review and editing; Bahman Afsari, Lijia Yu, Formal analysis; Stephania M Bezerra,

Christopher VandenBussche, Kazutoshi Fujita, Dilek Ertoy, Isabela W Cunha, Rachel Karchin, Chao-Yuan Huang, Resources, Investigation; Trinity J Bivalacqua, Ludmila Danilova, Chia-Tung Shun, Investigation; Arthur P Grollman, Conceptualization, Funding acquisition, Writing—review and editing; Thomas A Rosenquist, Writing—review and editing; Yeong-Shiau Pu, Resources, Supervision, Funding acquisition, Investigation, Writing—review and editing; Ralph H Hruban, Conceptualization, Visualization; Nickolas Papadopoulos, Ken W Kinzler, Bert Vogelstein, Kathleen G Dickman, George J Netto, Conceptualization, Formal analysis, Investigation, Visualization, Writing—original draft, Project administration, Writing—review and editing

### Author ORCIDs
Simeon U Springer https://orcid.org/0000-0002-2891-2111
Maria Del Carmen Rodriguez Pena http://orcid.org/0000-0002-3439-7013
Lu Li https://orcid.org/0000-0002-1920-4965
Christopher Douville http://orcid.org/0000-0002-2510-4151
Yuxuan Wang https://orcid.org/0000-0002-2932-6042
Joshua David Cohen http://orcid.org/0000-0003-1158-5668
Lijia Yu https://orcid.org/0000-0001-6735-9569
Bert Vogelstein https://orcid.org/0000-0003-0766-3854
Kathleen G Dickman https://orcid.org/0000-0003-1308-2992
George J Netto https://orcid.org/0000-0003-3915-9134

### Ethics
Human subjects: The study was approved by the Institutional Review Boards of Johns Hopkins Hospital and all other participating institutions. Proper material transfer agreements were obtained. Patients with a known history of malignancy other than urothelial cancer were excluded from the study. Additionally all UTUC patients provided informed consent using the consent form and study design reviewed and approved by the Institutional Review Boards at National Taiwan University and Stony Brook University. Consent to publish results was also obtained from all parties involved.

### Decision letter and Author response
Decision letter https://doi.org/10.7554/eLife.32143.033
Author response https://doi.org/10.7554/eLife.32143.034

## Additional files

### Supplementary files
• Supplementary file 1. Development of PCR-based assays to identify tumor-specific mutations in urinary cells.
DOI: https://doi.org/10.7554/eLife.32143.010

• Supplementary file 2. Demographic and clinical features of the BC early detection cohort.
DOI: https://doi.org/10.7554/eLife.32143.011

• Supplementary file 3. Mutations detected by Multiplex Assay in primary tumor tissues from the early detection cohort.
DOI: https://doi.org/10.7554/eLife.32143.012

• Supplementary file 4. Primers used to detect mutations in the 10-gene multiplex panel and TERT.
DOI: https://doi.org/10.7554/eLife.32143.013

• Supplementary file 5. Mutations detected by the 10-gene Multiplex Assay in urine samples from the early detection cohort.
DOI: https://doi.org/10.7554/eLife.32143.014

• Supplementray file 6. Mutations detected by the TERT Assay in urine samples from the early detection cohort.
DOI: https://doi.org/10.7554/eLife.32143.015

• Supplementary file 7. Chromosome arm gains and losses detected by the Aneuploidy Assay in urine samples from early detection cohort.
DOI: https://doi.org/10.7554/eLife.32143.016

• Supplementary file 8. Mutations detected by TERT Assay in primary tumor tissues from the early detection cohort.
DOI: https://doi.org/10.7554/eLife.32143.017

• Supplementary file 9. Individual clinical data for the UTUC cohort.
DOI: https://doi.org/10.7554/eLife.32143.018

• Supplementary file 10. Mutations detected in urinary cell DNA from UTUC patients by the 10-gene multiplex assay.
DOI: https://doi.org/10.7554/eLife.32143.019

• Supplementary file 11. TERT promoter mutations identified in urinary cell DNA from UTUC patients.
DOI: https://doi.org/10.7554/eLife.32143.020

• Supplementary file 12. Urinary cell DNA samples from UTUC patients that scored positive for aneuploidy.
DOI: https://doi.org/10.7554/eLife.32143.021

• Supplementary file 13. Comparison of copy number variations in matched tumor and urinary cell DNA samples from the UTUC cohort.
DOI: https://doi.org/10.7554/eLife.32143.022

• Supplementary file 14. Mutations detected in primary tumor DNA from UTUC patients by the 10-gene multiplex assay.
DOI: https://doi.org/10.7554/eLife.32143.023

• Supplementary file 15. TERT promoter mutations identified in primary tumor DNA from UTUC patients.
DOI: https://doi.org/10.7554/eLife.32143.024

• Supplementary file 16. Demographic and clinical features of the Surveillance Cohort.
DOI: https://doi.org/10.7554/eLife.32143.025

• Supplementary file 17. Mutations detected by the 10-gene Multiplex Assay on urine samples from the BC Surveillance cohort.
DOI: https://doi.org/10.7554/eLife.32143.026

• Supplementary file 18. Mutations detected by the TERT Assay on urine samples from the BC Surveillance cohort.
DOI: https://doi.org/10.7554/eLife.32143.027

• Supplementary file 19. Chromosome arm gains and losses detected by the Aneuploidy Assay in urine samples from the BC Surveillance cohort.
DOI: https://doi.org/10.7554/eLife.32143.028

• Supplementary file 20. Summary of the performance of Cytology vs.UroSEEK in both BC cohort tumors.
DOI: https://doi.org/10.7554/eLife.32143.029

• Supplementary file 21. Primers used for identity matching of tumor and urinary cell DNA samples from UTUC patients.
DOI: https://doi.org/10.7554/eLife.32143.030

• Transparent reporting form
DOI: https://doi.org/10.7554/eLife.32143.031

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
