## [Decision Letter]

Thank you for submitting your article "Non-invasive detection of bladder cancer through the analysis of driver gene mutations and aneuploidy" for consideration by *eLife*. Your article has been reviewed by 3 peer reviewers, and the evaluation has been overseen by Ross Levine as Reviewing Editor (and one of the reviewers) and Charles Sawyers as the Senior Editor.

The reviewers have discussed the reviews with one another and the Reviewing Editor has drafted this decision to help you prepare a revised submission.

In summary, in the two manuscripts the authors develop a non-invasive molecular test called UroSEEK to analyze DNA from urinary CELLS shed into the urine. UroSEEK combines three assessments including mutations from exons in 11 genes, TERT mutations, and aneuploidy. UroSEEK has the potential to augment the sensitivity of routine cytology for the detection of cancer in patients with bladder cancer and upper tract urothelial carcinomas. For the bladder study, they looked at two cohorts, one with hematuria and the second with previous bladder cancer history at risk for new bladder cancers. For the urothelial study, this included 56 subjects with urothelial cancer. For each cohort, they used the same 188 control urine specimens from healthy subjects as negative controls. The main conclusion from these studies is the same – urinary cell mutation testing has potential to detect urothelial and bladder cancers. As such, we think it is best to combine these papers into a single report, which would be of broad interest to the field.

Essential Revisions:

1) There is significant overlap between these two manuscripts, and reading these separately does not add value to either one's conclusion, especially in the same journal issue of *eLife*. The clinical and diagnostics audience for urothelial and bladder cancer is the same. While the clinical cohorts are different, the genes, methods, controls, and disease (based on their common mutation assessment with URoSEEK) are identical. The written methods are identical. The message is the same – urinary cell mutation testing has potential to detect urothelial and bladder cancers.

2) The issue of how this approach compares to cystoscopy is a major one and is not thoughtfully discussed. If both tests are similar in cost, and cystoscopy is more invasive but definitive, will patients and payers want to use an equally costly non-invasive test vs the more definitive test? How would the authors propose to use this as part of the diagnostic workup and can the impact be modeled in terms of costs, # procedures, etc.? Would add value.

3) The lack of specificity, particularly with respect to positive genetic data in 1) patients without cancer and 2) in patients who previously had cancer and did not recur, is a substantive issue. Can this be improved? Are there ways to deal with this issue in terms of proposing a diagnostic approach? On this note, the methods/assay used have not been thoroughly vetted for sensitivity, specificity, limits of detection and reproducibility for a clinical assay, typically done with serial dilutions and replicates. The authors indicate that one reason for false negatives was due to discordant mutations on the gene panel and the tumor. Another reason could be the assay's limits of detection. While this has not been completed, nonetheless, it does not significantly diminish the conclusion from these studies that urinary cell mutation testing could augment cytology.

4) Have the authors modeled whether including more, or less than 11 genes + TERT/aneuploidy, would impact performance of the test? Are there ways to improve the design to improve performance?

5) Why did the authors separate these assays especially that the three assays that make up UroSEEK can be combined in one NGS platform (examples include assays developed by Foundation Medicine, MSK-IMPACT, Oncomine panel, etc.) that can provide more comprehensive genetic analysis on many more genes including mutations and copy number alterations. Some of the currently available NSG platforms can also utilize cfDNA.

The list of genes in the assay, while includes some of the very commonly mutated genes in bladder cancer, it leaves out other genes that are both relevant and common including for example ARID1A, KDM6A, CDKN1A, STAG2, MLL2/MLL3, etc. Including such genes would very likely improve the assay.

---

## [Author Response]

Essential Revisions:

*1) There is significant overlap between these two manuscripts, and reading these separately does not add value to either one's conclusion, especially in the same journal issue of* eLife*. The clinical and diagnostics audience for urothelial and bladder cancer is the same. While the clinical cohorts are different, the genes, methods, controls, and disease (based on their common mutation assessment with URoSEEK) are identical. The written methods are identical. The message is the same – urinary cell mutation testing has potential to detect urothelial and bladder cancers.*

The two manuscripts have been integrated into a single paper as requested.

2) The issue of how this approach compares to cystoscopy is a major one and is not thoughtfully discussed. If both tests are similar in cost, and cystoscopy is more invasive but definitive, will patients and payers want to use an equally costly non-invasive test vs the more definitive test? How would the authors propose to use this as part of the diagnostic workup and can the impact be modeled in terms of costs, # procedures, etc.? Would add value.

This is an important point, and we appreciate the reviewer's bringing it up. Cystoscopy is definitely definitive but is invasive and always associated with pain and inconvenience and occasionally associated with side effects such as infection, bleeding, urethral stenosis, etc. The procedure also requires skilled professionals to perform it. As the great majority of patients in some screening settings (e.g., hematuria) do not have cancer, these issues are precisely those that inspired our study. It is widely agreed that a non-invasive test that could be used to screen patients in such settings could minimize the need for cystoscopy and its unintended consequences.

The cost of cystoscopy is up to $3200, not including the cost of the pathology of biopsies. The cost of ureteroscopy is similar. We envision that the cost of UroSEEK plus cytology would be <$750 when implemented in a commercial setting, using savings afforded by economy-of scale. We anticipate that >90% of patients with hematuria who would otherwise undergo cystoscopy could be spared this procedure on the basis of a negative UroSEEK test The entire cost savings, integrated over all patients, would therefore be considerable: for every 100 patients evaluated, up to $200,000 would be saved (assuming UroSEEK plus cytology in all patients plus cystoscopy in 5% of the patients who test positive with UroSEEK). This translates to many millions of dollars in the U.S. annually, and does not include the cost of follow-ups for the side effects of cystoscopy when performed. These issues are now briefly discussed at the end of the revised Discussion.

3) The lack of specificity, particularly with respect to positive genetic data in 1) patients without cancer and 2) in patients who previously had cancer and did not recur, is a substantive issue. Can this be improved? Are there ways to deal with this issue in terms of proposing a diagnostic approach?

We agree with the reviewers that specificity is important, but there are two kinds of specificity. Technical specificity can be assessed from the evaluation of the test in DNA from totally normal individuals, such as DNA from normal WBCs. The technical specificity of UroSEEK is very high, over 99%, as reported in the manuscript. Biological specificity is different and reflects the cohorts studied. In urines from normal individuals (without any hematuria, for example), the specificity is as high as in as in normal WBCs (>99%). As noted in the manuscript, though, this specificity is not as high in other cohorts. In the Discussion, this issue is discussed at length. For example, the 6.5% false positives in BC early detection cohort could be related to a lead time ahead of “clinical” detection. The follow-up period for many of patients was only one year, and cystoscopy was not performed in every patient as per current standards. The same issue could lead to apparently false positive results in the surveillance cohort, in which patients are at much higher risk for cancer than in the early detection cohorts. We will attempt to address such possibilities in future prospective studies that we are now planning where a longer follow up (e.g. 3 years minimum) should be helpful. False positives are not unique to our study; they have been observed in all other molecular assays for bladder cancer, including those that are FDA- approved. Even assuming that none of the false positives are due to as yet undetected cancers, the positive predictive values and negative predictive values in the early detection cohorts is considerably above what has been required for FDA-approval of diagnostic tests in general.

On this note, the methods/assay used have not been thoroughly vetted for sensitivity, specificity, limits of detection and reproducibility for a clinical assay, typically done with serial dilutions and replicates. The authors indicate that one reason for false negatives was due to discordant mutations on the gene panel and the tumor. Another reason could be the assay's limits of detection. While this has not been completed, nonetheless, it does not significantly diminish the conclusion from these studies that urinary cell mutation testing could augment cytology.

We appreciate the reviewers’ view that such detailed vetting is not critical for the conclusions made in this manuscript. But we would like to highlight several pieces of data that address this point. The reproducibility of the test has been exhaustively examined, because tests were scored positive *if and only if* an independent assay of DNA from the same urine sample was also positive. Moreover, the coefficients of variation of the duplicates were high, as noted in the table below (which can easily be recalculated by readers using the data recorded in the Supplementary files, wherein the results of each replicate are specified). We note that the CVs are relatively low even when the MAFs are low, meaning that there are only a few molecules present, helping to establish technical sensitivity (limit of sensitivity).

UroSEEK requires no complex molecular biological manipulations of DNA; it simply uses PCR of DNA purified from the urine. Thus, many of the issues that are faced by other next generation sequencing technologies, such as those involved in end-polishing of DNA, ligations, and captures, are avoided. The reproducibility and sensitivity of SafeSeqS, which is the basis for UroSEEK, has been previously documented in the original paper (Kinde et al., Detection and quantification of rare mutations with massively parallel sequencing. Proc Natl Acad Sci U S A, 108(23), 9530-9535).

Finally, we are in the process of licensing UroSEEK to a small start-up company that was founded by some of the senior authors of this study. As part of their due diligence and preparation for an upcoming clinical trial of UroSEEK based on the results reported in our manuscript, they have prepared a Clinical Laboratory Improvement Amendment (CLIA) report. Though the report cannot be made public, it provides additional evidence of the reproducibility of UroSEEK in a completely different laboratory.

CV (SD/Mean*100)Mutation Group (classified by MAF)All>10%>5%>2%>1%>0.5%<0.5%All16.558.399.5511.7013.1015.0235.53BC Surveillance18.008.729.6812.7013.4814.9532.23BC early detection15.868.6710.0911.6013.2615.1727.21UTUC cohort15.185.976.388.8211.1614.586.77

4) Have the authors modeled whether including more, or less than 11 genes + TERT/aneuploidy, would impact performance of the test? Are there ways to improve the design to improve performance?

This is an excellent question. The genes chosen for the analysis were based on a comprehensive evaluation of the literature on bladder cancer (insufficient data existed for UTUC, but prior publications indicate that the genetic alterations in upper and lower urinary tract cancers are similar). We modeled how additional genes would impact the assay by looking at the inclusion of other amplicons from the top 20 genes contributing to bladder cancer development listed in the COSMIC database. As shown in the figure in Supplementary file 1, the 59 amplicons chosen are above the asymptote in this graph, meaning that more amplicons would not substantially to sensitivity. More amplicons would however, increase the cost and indirectly decrease the sensitivity by decreasing specificity. The reason for this decrease in sensitivity is that the probabilistic models used to interpret rare mutations are rooted in multiple hypothesis testing; the more amplicons studied, the greater the possibility for false positives. The Bonferroni correction is an early example of an approach for dealing with this issue. The new figure illustrates that our choice of amplicons was reasonable. Moreover, the point in the graph (at 59 amplicons) shows that the expected sensitivity was actually realized in the primary bladder cancers that we studied, and was not just theoretical based on expectations from the COSMIC database.

We have included this graph as a new figure in Supplementary file 1 to illustrate this important concept.

5) Why did the authors separate these assays especially that the three assays that make up UroSEEK can be combined in one NGS platform (examples include assays developed by Foundation Medicine, MSK-IMPACT, Oncomine panel, etc.) that can provide more comprehensive genetic analysis on many more genes including mutations and copy number alterations. Some of the currently available NSG platforms can also utilize cfDNA.

As mentioned in the manuscript, *TERT* promoter region is very CG rich and difficult to amplify. This forced us to exclude it from the rest of the panel and sequence it alone. The Aneuploidy test requires a completely different amplification strategy due the large number of total regions that are assessed (~30,000).

We are keenly aware of the other tests cited by the reviewers because they are largely based on the strategy we described earlier, using endogenous rather than exogenous molecular barcodes (Kinde reference above). Though it is beyond the scope of our article to discuss their strengths and weaknesses, we note that these other tests are designed to detect relatively high fractions of mutant DNA, much higher than those we observed in many urine samples, at a much larger number of genomic positions. Additionally, they are far more expensive than UroSEEK and not well-suited for screening purposes. UroSEEK is unique so far in that it can assess a relatively small (but adequate) number of amplicons at very high depth and sensitivity at minimal cost.

The list of genes in the assay, while includes some of the very commonly mutated genes in bladder cancer, it leaves out other genes that are both relevant and common including for example ARID1A, KDM6A, CDKN1A, STAG2, MLL2/MLL3, etc. Including such genes would very likely improve the assay.

We again thank the reviewers for this suggestion, but note that the inclusion of these other amplicons would not substantially increase the sensitivity of the assay, as documented by the figure in Supplementary file 1 (and now included in the revised manuscript). We also note that cancers that do not harbor mutations in the 59 individual amplicons evaluated can still be aneuploid and can therefore be detected with the third component of UroSEEK, which detects aneuploidy.